Main Manuscript for
**Hygroscopicity of Isoprene-Derived Secondary Organic Aerosol Mixture Proxies:**
**Importance of Diffusion and Salting In Effects**
Nahin Ferdousi-Rokib[1*], N. Cazimir Armstrong[2], Stephanie Jacoby[3], Alana J. Dodero[4], Martin
Ahn[1], Ergine R. Remy[1], Zhenfa Zhang[2], Avram Gold[2], Joseph L. Woo[5], Yue Zhang[4], Jason D.
Surratt[2,6], Akua A. Asa-Awuku[1,3]
[1]Department of Chemical and Biomolecular Engineering, University of Maryland, College Park,
MD 20742, United States
[2]Department of Environmental Sciences and Engineering, University of North Carolina at
Chapel Hill, Chapel Hill, North Carolina 27599, United States
[3]Department of Chemistry and Biochemistry, University of Maryland, College Park, MD 20742,
United States
[4]Department of Atmospheric Sciences, Texas A&M University, College Station, Texas 77843,
United States
[5]Department of Chemical and Biomolecular Engineering, Lafayette College, Easton, PA 18042,
United States
[6]Department of Chemistry, University of North Carolina at Chapel Hill, Chapel Hill, North
Carolina 27599, United States
[*]Now at: Department of Environmental Health and Engineering, Whiting School of Engineering,
Johns Hopkins University, Baltimore, MD 21218, United States
Keywords: Hygroscopicity, Organic Aerosols, IEPOX, SOA, viscosity, AFM
Correspondence to: Nahin Ferdousi-Rokib (ferdousn19@gmail.com) and Akua A. Asa-Awuku
(asaawuku@umd.edu)

31

32

## Abstract

Isoprene-derived secondary organic aerosol (SOA) constituents, such as the 2-methyltetrols (2-MT) and 2-methyltetrol sulfates (2-MTS), have been readily detected in atmospheric aerosols ($PM_{2.5}$) and within mixtures containing ammonium sulfate (AS). Despite its prevalence, the water uptake of 2-MT, 2-MTS, and their mixtures are not well understood. In this study, we determine the physicochemical properties (e.g., surface activity, diffusivity, phase morphology) of synthesized 2-MT, 2-MTS samples, and their mixtures with AS. 2-MT and 2-MTS have been identified as surface-active and viscous. Thus, dynamic surface tension ($\sigma_{s/a}$) measurements were taken to determine organic diffusion coefficients ($D_s$). The droplet growth of organic/AS mixtures was measured under subsaturated conditions using a humidified tandem differential mobility analyzer (H-TDMA) at 88.2% RH ± 1.5%. Droplet activation was measured under supersaturated (> 100% RH) conditions using a cloud condensation nuclei counter (CCNC); supersaturation (*SS*) ranged from 0.3-1.4%. Hygroscopicity in both regimes were parameterized by the single hygroscopicity parameter $\kappa$.

This study demonstrates how diffusion and salting-in effects influence the water uptake of synthesized, isoprene-derived SOA mixtures. Results show that when mixed with AS, organic diffusion for 2-MTS/AS becomes an order of magnitude faster while 2-MT diffusivity remains unchanged. Both 2-MT/AS and 2-MTS aerosols present a plateau in subsaturated $\kappa$-values close to pure AS. However, under supersaturated conditions, 2-MTS/AS behaves ideally, well-mixed , and can be characterized by $\kappa$-Köhler theory. Isoprene-derived SOA like 2-MT and 2-MTS samples are ubiquitous, and thus, the impact from biogenic sources and its non-ideal thermodynamic properties must be considered in aerosol-cloud interactions.

## 1. Introduction

Fine aerosol particles (PM$_{2.5}$) suspended within our atmosphere are a major contributor to Earth's radiative forcing and uncertainties in global temperature projections (Intergovernmental Panel on Climate, 2023). Aerosol-cloud radiative forcing uncertainty is attributed to aerosols' ability to form and modify cloud properties, known as aerosol-cloud interactions or the "aerosol indirect effect" (Köhler, 1936; Twomey, 1959; Twomey, 1974; Albrecht, 1989; Intergovernmental Panel on Climate, 2023). An aerosol's ability to alter droplet formation is dependent on its hygroscopicity or water uptake behavior under supersaturated conditions (RH > 100%). In the presence of water vapor, aerosols present a surface for condensation; droplet activation depends on aerosol particle chemical composition and size (Seinfeld & Pandis, 1998; Petters & Kreidenweis, 2007). The aerosol droplets can reach a point of unstable and uncontrollable growth, thereby acting as cloud condensation nuclei (CCN) (Köhler, 1936; Seinfeld & Pandis, 1998).

Droplet models can apply Köhler theory to estimate aerosol droplet growth and CCN activity (Köhler, 1936). In traditional Köhler theory, it is assumed that all aerosol solutes instantaneously dissolve and contribute to water uptake (Petters & Kreidenweis, 2007). Aerosol hygroscopicity is thus parameterized by Köhler theory through the single hygroscopicity parameter $\kappa$; $\kappa$ of mixed composition is often estimated by the Zdanovskii-Stokes-Robinson (ZSR) mixing rule and it is assumed that an individual solute's contribution to hygroscopicity is scaled by its volume fraction (Petters & Kreidenweis, 2007). Thus, knowing aerosol composition is critical for understanding CCN formation. However, $\kappa$-Köhler predictions of aerosol CCN activity neglect solute physicochemical properties that may alter droplet growth. Previous studies have shown that droplet-altering properties may be present within aerosols, such as the presence of complex morphologies (e.g., inner core-outer layer), surface-activity, or salting in/salting out effects; as a result, discrepancies between experimentally-determined $\kappa$ and $\kappa$-Köhler predictions may occur (Asa-Awuku & Nenes, 2007; Bertram et al., 2011; Song et al., 2013; Prisle & Mølgaard, 2018; Riemer et al., 2019; Ott et al., 2020; Malek et al., 2023).

Field studies have observed the presence of internally mixed aerosols containing both inorganic and organic compounds (Saxena, 1995; Murphy et al., 1998; Pratt & Prather, 2010). Inorganic aerosols, primarily composed of salts like ammonium sulfate (AS) and sodium chloride have well-defined hygroscopic properties. The ionic behavior of inorganic compounds promotes instantaneous dissolution in water and contributes to CCN activation (Cziczo et al., 1997; Seinfeld, 2003; Rose et al., 2008; Laskina et al., 2015). However, fine organic aerosols (OA) pose a greater challenge to aerosol hygroscopicity predictions. OA constitute 20-50% of atmospheric fine aerosol mass and are diverse in composition. OA can be directly emitted into the atmosphere, referred to as primary organic aerosols (POA) (Kanakidou et al., 2005). POA can originate from anthropogenic (e.g., biomass burning and coal combustion) and biogenic (e.g., pollen) sources (Seinfeld & Pandis, 1998; Kanakidou et al., 2005). In addition to POA, secondary organic aerosol (SOA) can be formed through multigeneration gas-phase oxidation reactions of volatile organic compounds (VOCs) or multiphase reactions of semi-/low-volatility organic compounds (SVOCs/LVOCs) (Kanakidou et al., 2005). SOA is ubiquitous in the atmosphere, forming a major component of fine OA mass (Zhang et al., 2007; Srivastava et al., 2022). For example, a study by

Zhang et al. (2007) found that SOA contributed 65% to 95% of OA mass in urban and remote
regions. Furthermore, SOA have been readily detected in mixtures with inorganic components,
such as AS (Yang et al., 2009; Zhu et al., 2017); indeed, a study by Zhu et al. (2017) estimated
66% of SOA as being internally mixed with sulfate. Thus, in addition to understanding pure
organic compounds, it is important to also study organic-inorganic interactions.
Previous studies have determined that a significant contributor to SOA is the aqueous-phase
chemical processing of isoprene-derived oxidation products (Claeys et al., 2004; Kanakidou et al.,
2005). Isoprene is a VOC emitted from biogenic sources and is considered one of the most
abundant biogenic VOCs (BVOCs). Isoprene emissions have been estimated to be ~ 500 Tg C
year$^{-1}$, rivaling methane emissions (Guenther et al., 2012; Sindelarova et al., 2014). Under alkyl
peroxy radical (RO$_2$$^•$) + hydroperoxy radical (HO$_2$$^•$) dominant conditions, isoprene is
photochemically oxidized by gas-phase hydroxyl radicals ($^•$OH) to form large quantities of
isoprene-derived epoxydiols (IEPOX) (Paulot et al., 2009). IEPOX is then able to partition into
acidic sulfate-containing aerosol particles to produce isoprene-derived SOA (Surratt et al., 2010;
Lin et al., 2012; Gaston et al., 2014; Riva et al., 2019), which consists largely of 2-methyltetrols
(2-MT) and 2-methyltetrol sulfates (2-MTS).
Both 2-MT and 2-MTS were previously detected in atmospheric PM$_{2.5}$. For example, a study by
Claeys et al. (2004) found that 2-MT contributed 2% of organic carbon detected in PM$_{2.5}$ collected
from the Amazon rainforest. Additional field studies have also found that 2-MTS can contribute
0.3-16.5% of total organic carbon in both the Amazon rainforest and Southeast US (Chan et al.,
2010; Froyd et al., 2010; Hettiyadura et al., 2019; Riva et al., 2019; Chen et al., 2021; Hughes et
al., 2021). The formation of both compounds can also alter aerosol particle composition and phase
state (Zhang et al., 2019a; Zhang et al., 2019b). For example, 2-MT and 2-MTS have been
observed to be in a semisolid or glassy state in aerosol particles (Chen et al., 2023). Highly viscous
SOA can exist in a glassy state; SOA viscosities can range from $10^2$ to $10^{12}$ Pa·s for ultraviscous
liquids or $>10^{12}$ Pa·s for amorphous, extremely viscous compounds (Virtanen et al., 2010;
Renbaum-Wolff et al., 2013; Zhang et al., 2015). Viscosity can influence organic solute dissolution
in droplets by slowing diffusion through the aqueous phase (Renbaum-Wolff et al., 2013).  As a
result, slower organic diffusion rates can influence gas partitioning, particle shape, chemical aging,
multiphase reactions, and aerosol droplet growth (Riipinen et al., 2011; Shiraiwa & Seinfeld, 2012;
Zhang et al., 2015). Furthermore, studies incorporating SOA viscosity and phase state into larger,
global-scale models have observed changes to CCN and ice nuclei (IN) formation predictions
(Riipinen et al., 2011; Shiraiwa et al., 2017; Wolf et al., 2021). Thus, probing the viscosity and
resulting diffusion limitations may be necessary for understanding 2-MT and 2-MTS water uptake
properties (Chen et al., 2023).
Similar to other complex organic mixtures, the water uptake ability of isoprene-derived SOA can
be further complicated when mixed with inorganic components, such as AS. Previous studies have
observed the presence of internally-mixed SOA/AS aerosols in both the southeast US and Amazon;
in both regions a strong presence of 2-MT and 2-MTS has been observed (Chan et al., 2010; Froyd
et al., 2010; Bondy et al., 2018; Riva et al., 2019; Wu et al., 2019). The presence of inorganic salts
in aerosol mixtures can influence phase state based on organic physicochemical properties

(Topping, 2010; Ruehl et al., 2012; Ruehl et al., 2016; Malek et al., 2023). Inorganic compounds can result in water solubility-limited and/or surface-active organics partitioning to a separated phase (Ruehl et al., 2012; Ruehl et al., 2016; Freedman, 2017; Kang et al., 2020). As a result, the partitioned aerosols can exhibit a phase separated morphology (e.g., but not limited to Ruehl et al., 2012; Ruehl et al., 2016; Freedman, 2017; Kang et al., 2020; Malek et al., 2023). However, inorganic salts may also enhance organic dissolution, known as "salting in" (Riva et al., 2019). For instance, studies have observed increased diffusion in viscous SOA particles through the aqueous droplet phase in the presence of inorganic salts (Reid et al., 2018; Jeong et al., 2022; Sheldon et al., 2023). Increased diffusion is a result of salts disrupting the hydrogen bonding network between neighboring organic molecules (Reid et al., 2018; Jeong et al., 2022; Sheldon et al., 2023). Therefore, organic physicochemical properties (surface-activity, viscosity) of SOA, such as 2-MT and 2-MTS, must be better defined to better predict mixed SOA/AS aerosol CCN activity. To our knowledge there are no studies to date that investigate 2-MT and 2-MTS aerosol water uptake, water uptake of mixtures with AS, and the potential effect of physicochemical properties on CCN activity predictions.

In this study, we investigated the surface activity, diffusivity, droplet growth and water uptake of 2-MT, 2-MTS, and their mixtures with AS. 2-MT and 2-MTS surface tension values were experimentally determined. A previous study by Ekström et al. (2009) found 2-MT to be moderately surface-active. However, the surface activity of 2-MTS has not been characterized and potential organic surface tension depression in the presence of AS has not been explored for both organics. In tandem with surface tension measurements, this study estimated diffusion coefficients of both compounds to explore the effects of viscosity and diffusivity on aerosol water uptake. Aerosol $\kappa$-hygroscopicity for pure organic and organic-AS mixtures were experimentally determined under both subsaturated conditions (< 100% RH) and supersaturated (> 100% RH) conditions to observe both droplet growth and CCN activity, respectively. $\kappa$-hygroscopicity measurements were then compared to $\kappa$-Köhler hygroscopicity theory to evaluate the efficacy of traditional full dissolution and negligible viscosity assumptions in predicting the CCN activity of both compounds and their mixtures. Lastly, Atomic Force Microscopy (AFM) measurements on mixed particles were conducted to further understand particle morphology. The following work provides a comprehensive analysis of the wide range of physicochemical properties that may influence the droplet growth of 2-MT and 2-MTS mixed with AS.

**2. Experimental Methods**

2.1. Experimental Chemicals

For this study, ammonium sulfate (AS, $(NH_4)_2SO_4$; Thermo Fisher Scientific, >99.0%), was purchased and used without further purification. 2-methyltetrol (2-MT) and 2-methyltetrol sulfate (2-MTS) samples were synthesized using the published procedure of Cui et al. (2018). 2-MT was determined to be > 98% pure. The purity of 2-MTS was determined to be ~73 wt%, with remaining sample mass estimated to be 3 wt% AS and 24 wt% sodium methyl sulfate (SMS). It should be

noted that from hereon 2-MTS sample refers to prescribed synthesized mixture and subsequent calculations account for the estimated contributions of AS and SMS.

2.2. Surface Tension Measurements

The surface tension of 2-MT, 2-MTS, and their mixtures with AS was measured at atmospherically relevant aqueous phase concentrations. Due to the limited amounts of synthesized sample, mixed amounts were judiciously selected to mimic mixture ratios previously reported in the literature. Specifically, a study by Cope et al. (2021) found that 2-MT concentrations in the atmosphere reached an upper bound of 300 mM. Therefore, stock solutions of 300 mM 2-MT and 2-MTS were prepared using deionized (DI) water. Furthermore, it is assumed that surface tension measurements at dilutions higher than 300 mM are also relevant for droplet growth. A study Bain et al. (2023) found that aerosol surface tension can be approximated from surface tension measurements of bulk mixtures composed of < 100 mM organic component. Additionally, recent studies (Mikhailov et al., 2024; Ferdousi-Rokib et al., 2025) also support the application of more dilute concentration regimes to predict droplet growth. A recent study by Mikhailov et al. (2024) found that surface tension depression observed in bulk dilute surface tension measurements was reflective of aerosol properties. Ferdousi-Rokib et al. (2025) also found that salting out effects can be approximated in mixtures having < 100 mM organic component. Thus, in this work, the stock solutions were diluted to a 3-94 mM range; each stock solution and subsequent dilution concentrations are provided in Supplemental Tables S1-S5.

Droplet surface tension ($\sigma_{s/a}$) was measured using a pendant drop tensiometer with a modified profile analysis tensiometer (SINTERFACE Inc.); the experimental set up has been described in Fertil et al. (2025). Briefly, the pendant drop tensiometer generates a droplet of solution (< 10 μL) suspended from a 0.9-mm diameter needle (Beier et al., 2019; Fertil et al., 2025). Droplets remain suspended for 300 s to reach equilibrium; at each time step (~1 s), the droplet $\sigma_{s/a}$ was obtained from fitting the droplet curvature to the Young-Laplace Equation (Fordham & Freeth, 1948; Spelt, 1996; Padró et al., 2010). Surface tension measurements were run in triplicate; prior to each measurement, the tensiometer was flushed with DI water and ~ 2 mL of solution. Measurements were obtained at ambient room conditions, with temperature range of 20.2-22 °C and relative humidity range of 40-45 % RH.

As the droplet equilibrates, surface tension changes, which is attributed to the accumulation of solute diffusing to the droplet surface (Joos & Rillaerts, 1981; Eastoe et al., 1998; Chernyshev & Skliar, 2015). As the solute saturates the surface, surface tension reaches equilibrium (Ross, 1945). The accumulation of solute at the surface and resulting concentration gradient within the droplet can be described by Fick's Second Law:

$$\frac{\partial c}{\partial t} = D_s \frac{\partial^2 c}{\partial x^2}, \tag{1}$$

where concentration over time $\frac{\partial c}{\partial t}$ is proportional to the second derivative concentration over position $\frac{\partial^2 c}{\partial x^2}$ and the diffusion coefficient $D_s$ ($m^2\ s^{-1}$). The dynamic surface tension can be correlated with solute diffusion over time as (Joos & Rillaerts, 1981):

$$\sigma_t = \sigma_0 - 2RTC \left(\frac{D_s t}{\pi}\right)^{0.5}, \tag{2}$$

where $\sigma_0$ is the starting surface tension, $\sigma_t$ is the surface tension at specified time $t$, $R$ is the universal gas constant, $T$ is temperature, and $C$ is organic molar concentration. Here, evaporation effects are negligible during the short suspension times. Therefore, the organic molar concentration $C$ is equivalent to the droplet solution concentration as Eq. 2 can then be rearranged to solve for $D_s$ using dynamic surface tension measurements.

## 2.3. Aerosol Experimental Methods

### 2.3.1. Aerosol Generation

Solutions of 0.1 g L$^{-1}$ total solute (2-MT, 2-MTS, and mixtures with AS) were prepared using ultra-purified Millipore water (18 MΩ·cm). Mixtures compositions are provided in Table S6. Polydisperse aerosols were then generated by passing each aqueous solution through a constant output Collison Nebulizer (Atomizer, TSI 3076); the generated aerosols were then dried to < 5% RH using two silica gel dyers in series. Aerosols were then analyzed for their water uptake properties under sub- and supersaturated conditions. To determine aerosol phase morphology, atomic force microscopy (AFM) images were also obtained. In addition to water uptake and AFM measurements, organic density and shape factor were measured; for details on density and shape factor measurements, see Armstrong et al. (2025).

### 2.3.2 Water Uptake Measurements

A humidified tandem differential mobility analyzer (H-TDMA) measured droplet growth under subsaturated conditions. Dry, polydisperse aerosols were size selected at 100, 150, and 200 nm by an electrostatic classifier (DMA 1, TSI 3082; flow rate = 0.3 L min$^{-1}$) and humidified using a Nafion humidification line (PermaPure M.H. series); particles were humidified at 88.2% ± 1.5% RH. Selected dry diameters are often assumed to be spherical, thus having a shape factor ($\chi$) of 1 (DeCarlo et al., 2004). Aerodynamic aerosol classifier (AAC) shape factor measurements confirmed 2-MT and 2-MTS sphericity (Armstrong et al., 2025). The wet diameter ($D_w$) was measured using a second electrostatic classifier (DMA 2, TSI 3082; flow rate = 0.3 L min$^{-1}$); the ratio of $D_w$ to the dry-size selected diameter ($D_d$) is equal to the growth factor ($G_F$). The experimental set up is provided in Fig. S1. To calibrate the H-TDMA, a 0.1 g L$^{-1}$ solution of AS was aerosolized; dried AS aerosols were size selected at 100 and 150 nm. Dried AS aerosol $G_F$ and instrument RH was measured, with calibration measurements repeated multiple times as reported in Table S7. The experimental solutions were then aerosolized, and $G_F$ was obtained for each solution; $G_F$ is used to calculate the hygroscopicity parameter under subsaturated conditions, $\kappa_{H\text{-}TDMA}$. In addition to subsaturated conditions, water uptake was measured under supersaturated (SS) conditions using a CCNC-100 (Droplet Measurement Technologies); the experimental set up is provided in Fig. S2. The theory and operation of the CCNC has been previously described (Roberts & Nenes, 2005; Lance et al., 2006; Rose et al., 2008). The Scanning Mobility CCN Analysis (SMCA) protocol was used to measure droplet activation (Moore et al., 2010). Briefly, the dried polydisperse aerosols were passed through an electrostatic classifier (TSI 3080) in

scanning mode and charged; scanning mode operated from 8-352 nm for 135 s. The DMA operated
at a sheath-to-aerosol flow rate ratio of 10:1, and aerosol sample flow rate of 0.8 L min$^{-1}$. The
monodisperse, size-selected aerosol stream was then sampled by a condensation particle counter
(CPC, TSI 3776, flow rate = 0.3 L min$^{-1}$) and the CCNC-100 (flow rate = 0.5 L min$^{-1}$) in parallel.
The CPC counted the number concentration of dry particles at a given particle size (condensation
nuclei, $N_{CN}$). The CCNC exposed the particles to 0.3-1.4%SS and the number concentration of
particles activated ($N_{CCN}$) were measured. The instrument set up was calibrated using AS (Rose et
al., 2008) and the calibration data are provided in Table S8 and Fig. S3.
The CPC counted the number concentration of dry particles at a given particle size (condensation
nuclei, $N_{CN}$). The CCNC exposed the particles to 0.3-1.4%SS and the number concentration of
particles activated ($N_{CCN}$) were measured. The instrument set up was calibrated using AS (Rose et
al., 2008) and the calibration data are provided in Table S8 and Fig. S3.
CCN data of AS and experimental solutions were analyzed using the Python-based CCN Analysis
Toolkit (PyCAT 1.0) (Gohil, 2022; Gohil & Asa-Awuku, 2022). PyCAT is a Python version of
SMCA and is available on GitHub for public use. The analysis toolkit calculated the activation
ratio $N_{CCN}/N_{CN}$ for each dry particle size. The activation ratios were fitted using a sigmoid curve
and the critical diameter ($D_{p, 50}$) was found, at which ~50% of the dry particles activate. A charge
correction is applied in PyCAT using the multi-charge correction algorithm previously described
(Fuchs, 1963; Wiedensohler, 1988). The obtained critical diameter of each solution is then used to
calculate the single hygroscopicity parameter under supersaturated conditions, $\kappa_{CCN}$.
2.3.3. Atomic Force Microscopy (AFM) Morphology
Atomic force microscopy (AFM) measurements were utilized to characterize aerosol phase
morphology. 2-MTS, 2-MTS/AS, and 2-MT/AS particles were collected onto silicon substrates
(Silson Ltd) using a cascade impactor (Sioutas Cascade Impactor, flow rate = 9 L min$^{-1}$ and stored
at room temperature and relative humidity (40-50% RH) prior to analysis. Imaging followed the
procedure of Zhang et al. (2018). Briefly, particles were imaged in a 5 x 5 μm region using a
Dimension ICON® AFM (Bruker) in tapping mode with resonant frequency of 150 kHz and spring
constant of 5.4 N m$^{-1}$.
2.3.4 Viscosity and Diffusion Calculation
The viscosity and the diffusion coefficients of the 2-MT and 2-MTS aerosols were calculated using
a modified Vogel-Tammann-Fulcher (VTF) equation (DeRieux et al., 2018). The dry glass
transition temperature values were determined to be 226 K and 276 K from a previous study by
Zhang et al. (2019b). The Gordon-Taylor coefficient and the fragility coefficient were assigned as
2.5 and 20, respectively. The hygroscopicity values were used from the measurement of H-TDMA
of this study.

**3. Traditional $\kappa$-Köhler theory**
Traditionally, water uptake of aerosol particles has been calculated using $\kappa$-Köhler theory (Köhler,
1936; Petters & Kreidenweis, 2007). Köhler theory considers aerosol physicochemical properties
(e.g., solute density, molecular weight) to describe the equilibrium water vapor saturation ratio at
a droplet's surface ($S_{eq}$). The equilibrium relationship encompasses two competing effects. The
Kelvin effect describes the increase of water vapor saturation as a result of the curvature of the
droplet; the Kelvin effect is represented by droplet surface tension $\sigma_{s/a}$. The Raoult (solute) effect
competes by decreasing vapor pressure due to the presence of solute in the aqueous droplet; the
solute effect is represented by the water activity term, $a_w$ (Seinfeld & Pandis, 1998; Wex et al.,
2008). For compounds dissolved in water, water activity can be parameterized by the single
hygroscopicity parameter, $\kappa$, as follows (Petters & Kreidenweis, 2007; Sullivan et al., 2009):

$$\frac{1}{a_w} = 1 + \kappa \frac{V_s}{V_w}, \tag{3}$$

where $V_w$ and $V_s$ are the volume of water and dry solute, respectively. Therefore, the equilibrium
saturation ratio ($S_{eq}$) over the droplet is described as:

$$S_{eq} = \left(1 + \kappa \frac{D_d{}^3}{D_w{}^3 - D_d{}^3}\right)^{-1} exp\left(\frac{4\sigma_{s/a}M_w}{RT\rho_w D_w}\right), \tag{4}$$

where $\rho_w$ is the density of water, $M_w$ is the molecular weight of water, $R$ is the universal gas
constant and $T$ is the temperature.
$\kappa$ describes ability of an aerosol to uptake water assuming full dissolution, and can be calculated
from the intrinsic properties of the solute as $\kappa_{int}$ (Sullivan et al., 2009):

$$\kappa_{int} = \frac{\nu_s \rho_s M_w}{\rho_w M_s}, \tag{5}$$

where $M_s$ is the molecular weight of solute, $\nu_s$ is the van't Hoff factor, and $\rho_s$ is the density of the
solute; Armstrong et al. (2025) found 2-MT and 2-MTS density to be 1.4 g cm$^{-3}$ and 2.46 g cm$^{-3}$,
respectively. To estimate $\kappa$-hygroscopicity of aerosols containing more than one compound, the
Zdanovskii, Stokes, and Robinson (ZSR) mixing rule can be applied to estimate  (Petters &
Kreidenweis, 2007):

$$\kappa_{ZSR} = \sum_i \varepsilon_i \kappa_i, \tag{6}$$

where $\varepsilon_i$ is the volume fraction of the individual solute component, $i$.
Experimental data can be used to derive aerosol $\kappa$. Under subsaturated (< 100% RH) conditions,
$G_F$ is related to hygroscopicity as follows (Kreidenweis & Asa-Awuku, 2014):

$$\kappa_{H\text{-}TDMA} = \frac{\frac{(G_F^3 - 1)}{RH}}{exp\left(\frac{4\sigma_{s/a}M_w}{RT\rho_w D_d G_F}\right)} - G_F^3 + 1, \tag{7}$$

Where $\kappa_{H\text{-}TDMA}$ is subsaturated hygroscopicity and RH is the relative humidity of the H-TDMA
instrument as a decimal. Similarly, for supersaturated (>100% RH), the critical diameter correlates
to $\kappa$ as follows (Petters & Kreidenweis, 2007):

$$\kappa_{\mathrm{CCN}} = \frac{4\left(\frac{4\sigma_{s/a}M_{w}}{RT\rho_{w}}\right)^3}{27D_{p,50}^3 \ln^2 SS}. \tag{8}$$

Where $\kappa_{\mathrm{CCN}}$ is supersaturated hygroscopicity. It is assumed that droplet surface tension $\sigma_{s/a}$ is equivalent to that of the surface tension of water $\sim 72$ mN m$^{-1}$. Köhler theory also assumes that all solutes are well mixed within the aqueous phase. The Köhler/ZSR model does not account for potential viscosity and diffusivity limitations due to inorganic-organic mixing in the aqueous phase. Therefore, in this study, $\kappa$- Köhler values are predicted assuming both 2-MT and 2-MTS are well mixed within the aqueous phase and fully contribute to droplet growth. The applicability of these assumptions is discussed in the later sections. Additionally, a list of variable abbreviations is provided in Table S9.

## 4. Results

4.1. Surface Tension and Diffusion

*Organic Samples*

Dynamic pendant drop tensiometer measurements were taken for 2-MT and 2-MTS samples; measurements were performed by hanging droplets < 10 μL over a period of 300 s. The droplet curvature was measured every 1 s. Average surface tension values were obtained for 2-MT and 2-MTS when droplet surface tension values remained constant (at equilibrium) and are listed in Table S10 and shown in Fig. 1.

In the dilute bulk measurement regime, 2-MT sample (Fig. 1, orange squares) and 2-MTS sample (Fig. 1, purple closed circles) $\sigma_{s/a}$ values are close to pure water ($\sim 72$ mN m$^{-1}$, Fig. 1, blue dashed line). For solutions < 53 mM organic concentration, 2-MT and 2-MTS samples exhibit little to no surface-activity. Surface-activity is similar to the dilute surface tension of pure AS, a non-surface-active compound, which remains $\sim 72$ mN m$^{-1}$(Fig. 1, red circles, Pruppacher et al., 1997). However, for organic solutions > 53 mM, minimal surface tension depression is observed with $\sigma_{s/a}$ values between $\sim$68–70 mN m$^{-1}$ (Fig. 1 and Table S10); in comparison, AS surface tension increases with concentration, as observed in Fig. 1 and with previous studies (namely, Pruppacher et al., 1997; Hyvärinen et al., 2005; Mikhailov et al., 2024). Therefore, both synthesized 2-MT and 2-MTS sample mixtures can be classified as weakly surface-active. A previous study by Riva et al. (2019) observed greater surface tension depression for IEPOX SOA/sulfate mixtures. In particular, enhanced surface tension depression was attributed to organic partitioning and formation of 2-MT and 2-MTS oligomers (Riva et al., 2019).

In comparison to the surface tension of other short-chained particulate organosulfates, such as sodium ethyl sulfate (Fig.1, black triangles) and sodium methyl sulfate (Fig. 1, grey triangles), 2-MT and 2-MTS have lower dilute surface values (Peng et al., 2021). However, similar to other surface-active organosulfates (sodium ethyl sulfate and sodium octyl sulfate), neither 2-MT sample and 2-MTS sample surface tension significantly depress aerosol surface tension (Table S11 and S14). For example, Mikhailov et al. (2024) observed surface tension depression as low as $\sim$

56 mN m$^{-1}$ for dilute D-glucose/AS mixtures.  Furthermore, moderately surface-active compounds, such as 2-methylglutaric acid (2-MGA, Fig. 1, green squares) and sodium octyl sulfate (Fig. 1, grey diamonds) exhibit surface tension depression in the range of $\sim$ 64-68 mN m$^{-1}$ for concentrations $\leq$ 22 mM (Tables S13-S14).   Additionally, stronger surface-active organics (surfactants), such as sodium dodecyl sulfate (SDS) show surface tension at the droplet surface can be depressed in the dilute regime. SDS reaches $\sigma_{s/a}$ of $\sim$ 39 mN m$^{-1}$ at 9 mM organic (Fig. 1 and Table S15). Sodium octyl sulfate, SDS, and 2-MGA present noticeable surface tension depression in the dilute bulk measurement regime (Fig. 1) that affect aerosol properties (Vepsäläinen et al., 2023; Zhang et al., 2023; Kleinheins et al., 2025). However, in comparison to previously studied organics, 2-MT and 2-MTS $\sigma_{s/a}$ samples remain close to pure water in the dilute

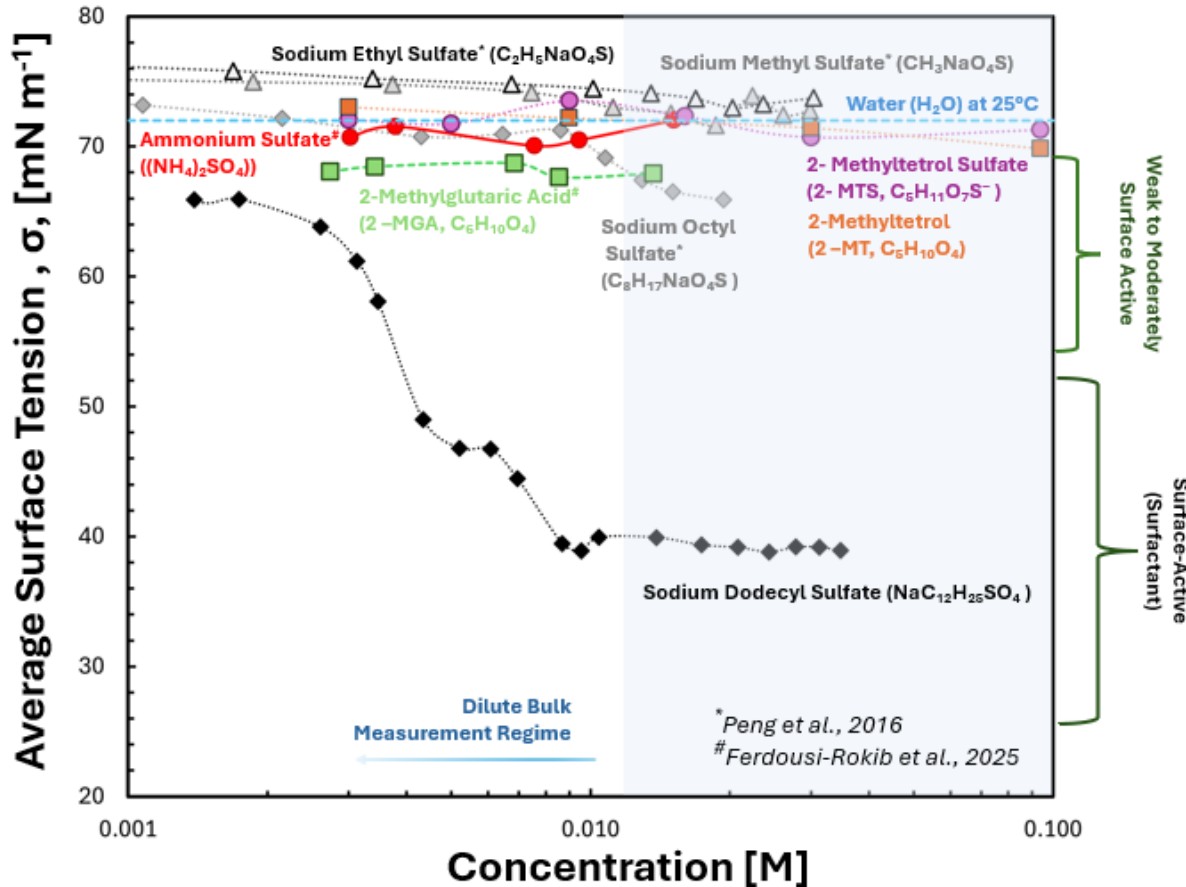

**Figure 1.** Experimental average surface tension $\sigma_{s/a}$ values of compounds as a function of concentration. Average equilibrium surface tension of synthesized 2-MT (>98 wt% purity) and 2-MTS (~73 wt% purity) samples are shown as closed orange squares and closed purple circles, respectively. The surface tension of the organosulfates, including sodium ethyl sulfate (black open triangles), sodium methyl sulfate (grey closed triangles), and sodium octyl sulfate (grey closed diamonds) were obtained by Peng et al. (2016). 2-methylglutaric acid (green closed squares) and ammonium sulfate (red closed circles) $\sigma_{s/a}$ were obtained from Ferdousi-Rokib et al., 2025 (in review). Sodium dodecyl sulfate (SDS) $\sigma_{s/a}$ is shown as black diamonds. Pure water $\sigma_{s/a}$ at 25°C ($\sim$ 72 mN m$^{-1}$) is represented as a dashed blue line. Compounds can be categorized as weak to moderately surface active (65-65 mN m$^{-1}$) or surface-active (surfactants, < 65 mN m$^{-1}$) for compounds that can depress surface tension below that of pure water. Bain et al 2023 consider the dilute bulk measurement regime to be less than 100mM.

bulk regime (Fig. 1). Previous studies by Bain et al. (2023) and Werner et al. (2025) emphasize the role of surface area-to-volume ratio dictating aerosol surface tension. Specifically, aerosol surface tension values are best represented by surface tension measurements of the organic in bulk solutions < 100 mM (Bain et al., 2023; Ferdousi-Rokib et al., 2025; Werner et al., 2025). Thus, 2-MT and 2-MTS sample mixture surface activity is negligible for droplet activation as both dilute organic $\sigma_{s/a}$ is close to that of pure water ($\sim$72 mN m$^{-1}$).

It should be reiterated and noted that the synthesized 2-MTS sample is 73% pure 2-MTS and is likely mixed with AS and SMS. Both SMS and AS (Fig.1, red circles; Table S16) have surface tension values, > 72 mN m$^{-1}$ in the dilute regime. However, despite the presence of impurities in the mixture, synthesized 2-MTS sample mixture surface tension reaches values $\sim$ 68 mN m$^{-1}$. Therefore, the presence of these impurities may counteract possible further surface tension depression exhibited by pure 2-MTS. Future work can better probe surface tension of the pure organic 2-MTS and effects of SMS by applying a multicomponent surface tension model (e.g., multicomponent models of Topping et al., 2007) to dynamic surface tension measurements.

Both 2-MT and 2-MTS are considered viscous compounds and may diffuse slowly through the measured droplets (Reid et al., 2018; Zhang et al., 2019a; Chen et al., 2023). As a result, equilibrium surface tension is reached after a period of time, $t$. The rate of diffusion of the organic through water, also known as the diffusion coefficient $D_s$, can be calculated from dynamic surface tension measurements (Eq. 1-2). Diffusion coefficient values for synthesized 2-MT and 2-MTS samples range between $10^{-9}$ to $10^{-11}$ m$^2$ s$^{-1}$, with diffusion slowing with increasing sample concentration. Specifically, $D_s$ for the 2-MT and 2-MTS samples are estimated to be $10^{-9}$ to $10^{-11}$ m$^2$ s$^{-1}$ and $10^{-9}$ to $10^{-10}$ m$^2$ s$^{-1}$, respectively (Table S17). Additionally, the viscosity-based diffusion coefficient was calculated and shown in Table S19. 2-MT and 2-MTS diffusion rates are comparable to rates observed for other previously investigated viscous components in aqueous solution (Curry et al., 2018; Tandon et al., 2019). For example, methylglyoxal, a known viscous component, has an aqueous phase diffusion rate $\sim$ $10^{-9}$ m$^{-2}$ s$^{-1}$ (Curry et al., 2018). In addition to the diffusion coefficients in aqueous solution, a study by Chenyakin et al. (2017) average diffusion coefficients between $10^{-13}$ and $10^{-14}$ m$^2$ s$^{-1}$ for organic molecules in a sucrose-water proxy for SOA. A study by Renbaum-Wolff et al. (2013) reported diffusion coefficients ranging from $10^{-13}$ and $10^{-15}$ m$^2$ s$^{-1}$ for $\alpha$-pinene-derived SOA between 70-90% RH. Indeed, 2-MT and 2-MTS have been previously observed to be highly viscous, resulting in slow diffusivity (Wang et al., 2011; Chenyakin et al., 2017; Tandon et al., 2019; Zhang et al., 2019a; Chen et al., 2023). Furthermore, at higher viscosity and lower diffusion rates, the diffusion of solute molecules fails to follow the Stokes-Einstein relationship describing the self-diffusion of solute molecules through a liquid phase (Einstein, 1905; Chenyakin et al., 2017; Tandon et al., 2019). For viscous material, such as 2-MT and 2-MTS sample, diffusion in water is self-limited (Chenyakin et al., 2017). Slow diffusion correlates with the longer time scales needed to reach equilibrium surface tension for more concentrated sample solutions; the solute molecules are limited in their ability to accumulate to the surface; thus, time is an important factor in the surface tension measurements. This effect is more prominent in 2-MT than 2-MTS sample, as evident in its slower diffusion rates for concentrations >30 mM (Table S17).

Previous studies have observed that inorganic compounds, such as AS, mixed with organics can
enhance surface tension effects (Topping, 2010; El Haber et al., 2023). Additionally, AS can result
in the partitioning of organics to the to the surface (i.e., the movement of organics to the surface is
commonly referred to as salting-out). To determine if partitioning effects are present in organic/AS
mixtures, synthesized 2-MT and 2-MTS samples were mixed with 500 mM AS and dynamic
surface tension measurements were taken; mixture dynamic surface tension measurements are
shown in Fig. 2. Average mixed surface tension values are listed in Table S10.

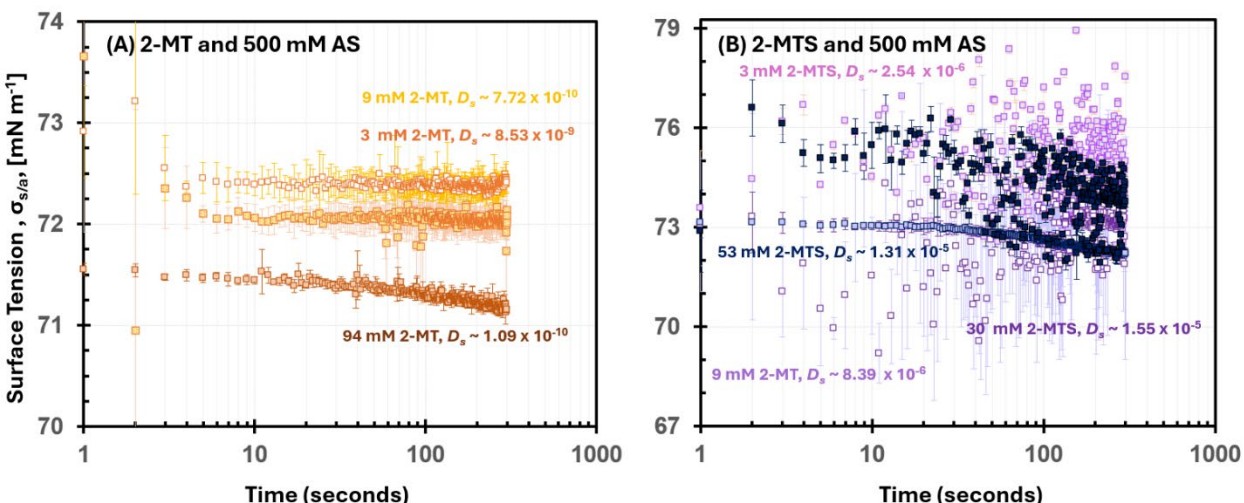

**Figure 2.** Dynamic $\sigma_{s/a}$ measurements for (A) 3-94 mM 2-MT sample/500 mM AS and (B) 3-53 mM 2-MTS sample/500 mM AS mixtures. Dynamic $\sigma_{s/a}$ was recorded over a duration of 300 seconds. The 2-MTS sample mixtures likely contain additional contibutions of AS (3 wt%) and SMS (24 wt%), which may further influence dynamic surface tension measurements and overall sample diffusivity.

For mixtures of 3-9 mM 2-MT and 500 mM AS, surface tension remains stable ~ 75 mN m$^{-1}$ and
is higher than 2-MT (>98 wt% purity) solution surface tension alone (Fig. 2A). Higher $\sigma_{s/a}$ values
indicate a lack of salting out effects and organic surface partitioning; previous surface tension
studies of organic/AS mixtures observed salting out effects through lower $\sigma_{s/a}$ values in comparison
to pure organic solutions (Ferdousi-Rokib et al., 2025 (in review)). Thus, for 3-9 mM 2-MT with
500 mM AS mixtures, organic partitioning is not enhanced, and the droplet surface tension aligns
with pure AS $\sigma_{s/a}$ (Fig.1. and Table S16). When organic concentration in the mixture is increased
to 94 mM, a stronger time dependence for surface tension is observed (Fig. 2A); an equilibrium
surface tension of ~71.2 mN m$^{-1}$ is reached at ~300 s. This lower surface tension for 94 mM 2-MT
with 500 mM AS compared to the previous 2-MT/AS mixture correlates with the higher
concentration of organic in solution. However, the longer equilibrium time is indicative of a slow
solute diffusion in the droplet.
Previous studies have observed diffusion effects within dynamic surface tension measurements
and estimated solute diffusion (Eastoe et al., 1998; Bain et al., 2024). To determine organic
diffusion within AS mixtures, the $D_s$ was calculated using Eqs. 1-2. For 2-MT/AS mixtures, $D_s$
ranged from $10^{-9}$ to $10^{-11}$, with diffusion slowing as organic concentration increases (Fig. 2A, Table
S17). 2-MT organic diffusion in AS mixtures is similar to that of the organic 2-MT solution (with
> 98% purity) $D_s$ values. As a result, 2-MT organic diffusion remains relatively unaffected in the
presence of AS. The organic 2-MT molecules do not diffuse fast enough to fully accumulate at the
surface and substantially lower surface tension.
Similar to 2-MT/AS mixtures, 2-MTS/AS mixture surface tension was higher than 2-MTS sample
solution surface tension alone. 2-MTS/AS mixture $\sigma_{s/a}$ values ranged from ~ 72.5 to 75 mN m$^{-1}$
and remain close to surface tension values of pure AS. Furthermore, $\sigma_{s/a}$ values remain constant as
the 2-MTS organic concentration increases from 3 to 53 mM; the minimal correlation between
organic concentration and surface tension implies that AS dominates droplet surface tension at the
surface-air interface. In addition to being stable across organic concentrations. 2-MTS/AS $\sigma_{s/a}$
reaches equilibrium faster than 2-MT/AS; equilibrium is achieved across the mixtures at < 100 s
(Fig. 2B). Indeed, based on the dynamic surface tension measurements, $D_s$ for 2-MTS within AS
mixtures remains ~$10^{-9}$, indicating slightly faster organic diffusivity through the droplet than 2-
MT (Table S17).  In the presence of AS, $D_s$ increases by an order of magnitude. This suggests the
presence of AS increases solubility and dispersion of 2-MTS molecules through the droplet, (Prisle
et al., 2010; Toivola et al., 2017). A similar phenomenon has been observed in glyoxal/AS mixtures
as the presence of the inorganic compound improves dissolution of the organic (Kampf et al.,
2013). Therefore, the higher 2-MTS/AS surface tension values and diffusivity indicate that the
organic is well dispersed within the droplet, but AS dominates droplet surface tension properties.
Both 2-MT and 2-MTS present complex viscous properties that may affect droplet phase and
potentially change in the presence of inorganic compounds, such as AS. It is important to note that
for 2-MTS, the remaining sample mass also contains SMS, which may further influence the
estimated diffusion rates (Vignes, 1966; Wallace et al., 2021). Diffusion coefficients within
aerosols may be sensitive to mixture ratio, as observed by Wallace et al. (2021). Thus, the presence
of SMS may affect the 2-MTS sample/AS diffusion rates observed in this study. Future work
should explore the influence of SMS on viscous organic diffusivity by applying this study's
methodology to a range of 2-MTS sample/SMS mixtures with 2-MTS contribution greater than 73
wt%. Ultimately, diffusion effects were observed through dynamic surface tension measurements
and may influence 2-MT, 2-MTS, and AS-mixed aerosol water uptake properties. Therefore,
additional diffusion effects on synthesized organic and organic/AS aerosol mixtures were probed
through the lens of water uptake measurements.

## 4.2. Water Uptake Measurements
In addition to the previous measurements, the droplet growth of 2-MT, 2-MTS samples, and their
respective AS mixtures were measured; hygroscopicity was estimated under both subsaturated and
supersaturated conditions. Mixtures were varied by sample wt% (Table S21); organic wt% of 2-
MTS is estimated by accounting for impurities present in the sample and their respective properties
(e.g., density, hygroscopicity, Table S20). The adjusted mass wt% for 2-MTS/AS mixtures are
listed in Table S21. For subsaturated hygroscopicity, the H-TDMA instrument setup was used to
measure $G_F$ for all experimental solutions at 88.2% RH. Experimental growth factor values for 2-
MT/AS and 2-MTS/AS mixtures are listed in Tables S20-S21. For supersaturated hygroscopicity,
the CCNC instrument setup was used to obtain experimental $D_{p,50}$ values across multiple
supersaturation conditions (0.31, 0.43, 0.65, 0.88, 1.10, 1.32, and 1.54 % SS); the critical diameter
values for 2-MT/AS and 2-MTS/AS mixtures are listed in Tables S22-S23. For 100 wt% 2-MTS
hygroscopicity, impurity (SMS and additional AS) hygroscopicity are accounted for by applying
ZSR mixing rule (Eq. 6) to solve for pure organic hygroscopicity; SMS $\kappa$ was assumed to be ~0.459
based on Peng et al. (2021).
Under subsaturated conditions, both 2-MT and 2-MTS are moderately hygroscopic, with $\kappa_{H\text{-}TDMA}$
values of 0.103 and 0.276, respectively (Fig. 3A). For 2-MT/AS (Fig. 3A, orange open squares)
and 2-MTS/AS (Fig. 3A, purple open circles) aerosol mixtures, subsaturated hygroscopicity values
are similar. For 2-MT/AS mixtures ≤ 45 wt% organic, $\kappa$ values plateau close to pure AS ($\kappa_{int}$ =
0.61) at a $\kappa_{H\text{-}TDMA}$ ~ 0.56. For mixtures > 45 wt% organic, both 2-MT and 2-MTS exhibit lower $\kappa_{H\text{-}}$

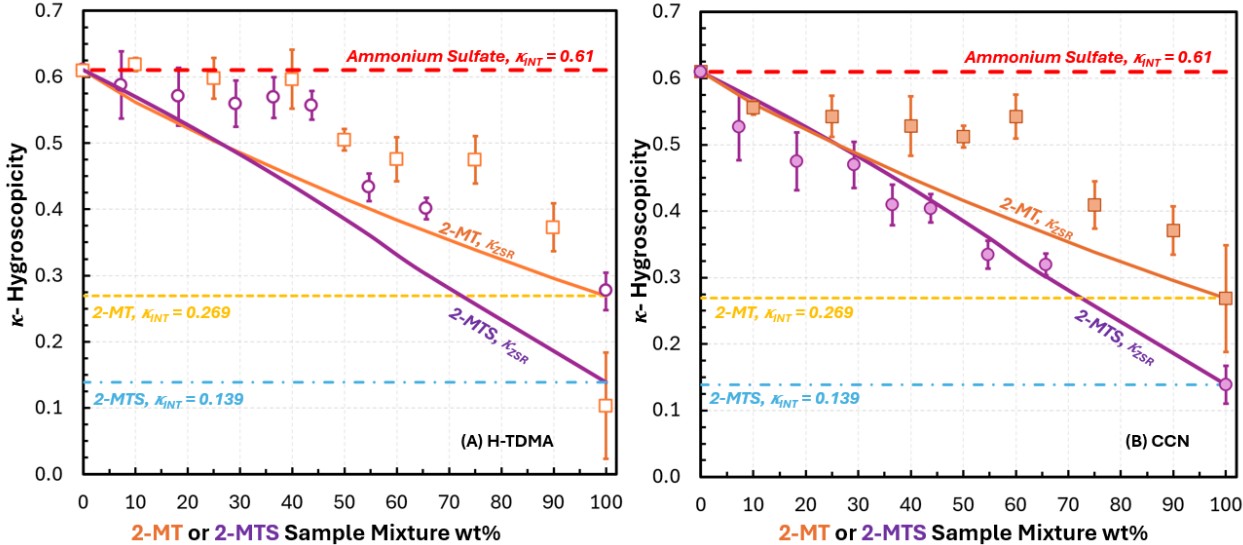

**Figure 3.** Experimental $\kappa$-hygroscopicity measurements derived from (A) H-TDMA measurements and (B)
CCNC measurements. 2-MTS/AS sample mixture wt% was adjusted based on the presence of AS and SMS
impurities (Table S21).  Subsaturated hygroscopicity ($\kappa_{H\text{-}TDMA}$) of 2-MT/AS and 2-MTS /AS sample
mixtures are represented as open orange squares and open purple circles, respectively. Supersaturated
hygroscopicity ($\kappa_{CCN}$) for 2-MT/AS and 2-MTS sample/AS mixtures are represented as orange squares and
purple circles, respectively. For 100 wt% 2-MTS, $\kappa$ values were adjusted to account for impurities by
applying mixing rule, assuming an SMS $\kappa$ of ~0.459 and AS $\kappa$ ~ 0.61 (Eq. 6, Table S20).  $\kappa$-Köhler theory
($\kappa_{ZSR}$) was used to predict hygroscopicity of 2-MT/AS (solid orange line) and 2-MTS sample/AS (solid
purple line) via Eq. 6. Organic $\kappa_{int}$ was determined from 100 wt% $\kappa_{CCN}$. 2-MT $\kappa_{int}$ (yellow dashed line) was
determined to be 0.269. 2-MTS $\kappa_{int}$ (blue dashed line) was determined to be 0.139.

$_{TDMA}$ values, ranging from 0.103-0.505 for 2-MT/AS mixtures and 0.276–0.433 for 2-MTS/AS
mixtures. Previous studies by Malek et al. (2023) and Ferdousi-Rokib et al. (2025)  have observed
a plateau in hygroscopicity for AS-dominated organic mixtures prior to a decrease in $\kappa$ due to the
presence of phase separated morphology; as a result of phase separation, the inorganic AS remains
dissolved in the aqueous phase and drives hygroscopicity (Malek et al., 2023). After a threshold
composition is reached (45 wt% organic), more organic solute contributes to the aqueous phase
and thus hygroscopicity is lowered.

Under supersaturated conditions, 2-MT and 2-MTS samples remain moderately hygroscopic, with $\kappa_{CCN}$ being 0.269 and 0.139, respectively. For 2-MT/AS sample mixtures (Fig. 3B, closed orange squares), supersaturated $\kappa$ mimics the same trend as subsaturated 2-MT/AS $\kappa$; for mixtures ≤ 60 wt% 2-MT, $\kappa_{CCN}$ also shows a plateau at ~ 0.53 and then decreases with increased organic aerosol composition. In comparison, the 2-MTS/AS sample mixtures (Fig. 3B, purple circles) present a linear hygroscopic trend; as organic wt% increases, $\kappa_{CCN}$ drops in a linear fashion resembling ideal mixing and volume additivity (Petters & Kreidenweis, 2007). Indeed, 2-MTS/AS $\kappa_{CCN}$ correlates with the hygroscopicity trend predicted by $\kappa_{ZSR}$ values (Eqs. 11-12) (Fig. 3B, purple line). 2-MTS/AS supersaturated hygroscopicity agrees well with original Köhler theory ($R^2 = 0.972$, Table S26), suggesting full 2-MTS dissolution and contribution to water uptake. By contrast, 2-MT/AS mixtures do not agree with $\kappa$-Köhler theory ($R^2 = 0.787$, Table S26), with the greatest discrepancy observed in the region between the $\kappa$ experimental plateau and $\kappa_{ZSR}$ (Fig. 3, orange line); additionally, subsaturated 2-MTS/AS mixtures deviate from $\kappa_{ZSR}$ during the initial hygroscopic plateau (Fig. 3A, purple line). Thus, for 2-MT/AS mixtures and subsaturated 2-MTS/AS aerosols, the ideal volume additive mixing rule does not apply. This can once again be attributed to limitations to organic dissolution into the aqueous phase (Malek et al., 2023). For 2-MTS/AS sample mixtures, both subsaturated and supersaturated hygroscopic trends may be further impacted by the presence by SMS. The contributions of AS and SMS hygroscopicity are accounted for 2-MTS sample mixture. $\kappa$ was estimated using ZSR mixing rule, which assumes ideal interactions between SMS, AS, and 2-MTS. However, non-idealities (e.g., phase separation, salting in) may result in SMS having a greater influence on hygroscopicity and can be the focus of future exploration.

In addition to non-ideal hygroscopic trends, it is noted that overall, $\kappa_{CCN}$ values remain lower than $\kappa_{H\text{-}TDMA}$ values for both 2-MT/AS and 2-MTS/AS sample mixtures, contrary to the usual trend of $\kappa_{CCN} > \kappa_{H\text{-}TDMA}$ (Petters & Kreidenweis, 2007). The observed difference suggests greater organic dissolution and contribution to hygroscopicity in the supersaturated regime compared to subsaturated conditions. This suggests potential viscosity and diffusion limitations on hygroscopicity as RH transitions from sub- to supersaturated. Indeed, the viscosity of the 2-MT and 2-MTS changes under different conditions. Both compounds remain in the semi-solid phase state before entering the CCNC, and behave like liquids in the H-TDMA, as shown in Table S18. Additionally, Asa-Awuku and Nenes (2007) report diffusivity limitation effects on aerosol water uptake for compounds with $D_s$ values ≤ 2.5 x $10^{-10}$, well within the range of $D_s$ values for 2-MT, and 2-MT sample/AS. Water uptake was shown to be driven by the viscous organic phase slowly diffusing into the aqueous phase (Asa-Awuku & Nenes, 2007). Thus, it is believed that both 2-MT and 2-MTS slowly dissolve and phase separate to form a viscous phase under subsaturated conditions, corresponding to slow diffusion coefficients. AS is an inorganic compound that is assumed to instantaneously dissolve into the aqueous phase and thus drives hygroscopicity when the droplet is phase separated, such as for 2-MT/AS mixtures (Fig. 2). However, lower $\kappa$ values at supersaturated conditions can be attributed to higher water content; previous studies have found greater water content correlating with reduced viscosity due to a plasticizing effect and resulting in enhanced organic mixing (O'Meara et al., 2016; Reid et al., 2018; Jeong et al., 2022). Thus, the organic viscous phase may experience "cracking" and greater movement of organic molecules through the aqueous phase (Tandon et al., 2019). Therefore, phase behavior of the organic can

have a strong influence on aerosol water uptake. Additionally, the non-ideal hygroscopic behavior
of 2-MT/AS and subsaturated 2-MTS/AS mixtures versus the ideal hygroscopic behavior of
supersaturated 2-MTS/AS aerosols can be probed through imaging of the aerosol mixture phase
behavior.

4.3. Phase Morphology
To further understand the phase state and morphology of 2-MT and 2-MTS sample mixtures with
AS, AFM images were taken at varied organic wt% (Fig. 4). Dried synthesized 2-MTS presents
itself as a viscous, spherical particle, indicated by its smooth surface (Fig. S4); this agrees with
both shape factor measurement of ~1 (Armstrong et al., 2025 (2025)) and diffusion coefficient
values. As inorganic AS is mixed with 2-MTS sample, phase behavior changes. At 10 wt% 2-MTS
sample (Fig. 4B), particles exhibit an engulfed core-shell morphology. A previous study by Cooke
et al. (2022) observed a similar core-shell morphology for AS-seeded IEPOX-derived SOA
particles; the study observed an organic shell, while the inorganic salt was observed to be present
in the shell as well as within an aqueous core (Cooke et al., 2022).  With AS dispersed on the outer
shell as well as being present in an aqueous core, the inorganic salt in the shell will likely easily
dissolve during water uptake and drive hygroscopicity, consistent with the results as observed in
subsaturated hygroscopicity measurements. However, AS within the shell may introduce

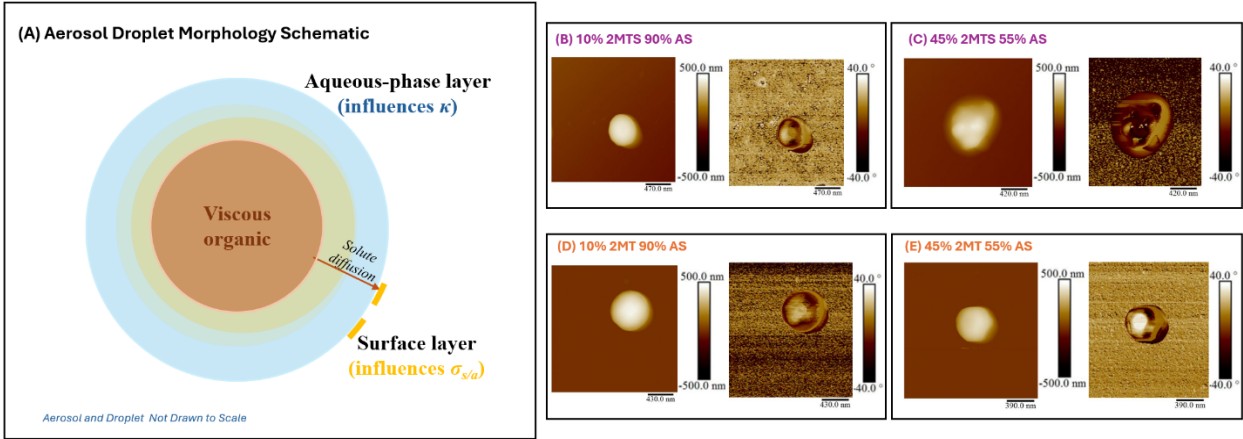

**Figure 4.** (A) Schematic depicting aerosol droplet composed of a viscous organic core and aqueous phase layer and AFM images of (B) 10 wt% 2-MTS – 90 wt% AS (C) 45 wt% 2-MTS – 55 wt% AS (D) 10 wt% 2-MT – 90 wt% AS and (E) 45 wt% 2-MT – 90 wt% AS. AFM results depict vertical particle height (left) and phase morphology (right).

roughness in the outer edge which can promote "cracking" in the organic phase, which can result
in full dissolution in the presence of higher water content and ideal mixing (Tandon et al., 2019).
As 2-MTS is increased to 45 wt%, the particle morphology shows greater inorganic phase
dispersion, with AS protruding through the viscous organic phase (Fig. 4C). The visualized
morphology and phase state of the particle agrees with behavior inferred from water-uptake and
droplet measurements (Sect. 4.2). In particular, ~45 wt% is the observed threshold for the plateau
in 2-MTS/AS $\kappa_{\text{H-TDMA}}$ values, prior to a linear decrease in $\kappa_{\text{H-TDMA}}$ values. The dispersion of AS
disrupts the organic network within the viscous phase, giving rise to the observed roughness and
promoting the salting in of 2-MTS. This phenomenon agrees with the results of previously
published literature that show viscous organics mixed with AS; specifically, laboratory-generated
SOA-AS and citric acid-AS mixtures (Saukko et al., 2012; Abramson et al., 2013). Previous
studies have also observed increased diffusion within viscous SOA particles via a disruption of the
hydrogen bonding network between the organic molecules that can promote solute movement in
the droplet (Reid et al., 2018; Jeong et al., 2022; Sheldon et al., 2023). For this reason, it is likely
that greater organic diffusion occurs above 45 wt% organic, resulting in decreasing $\kappa_{\text{H-TDMA}}$ values.
Furthermore, the well dispersed AFM morphology is indicative of ideal mixing under
supersaturated conditions, thereby agreeing with $\kappa$-Köhler theory of droplet growth.
In comparison, 2-MT mixtures present an engulfed core-shell morphology from 10 to 45 wt%
organic (Fig. 4D-E). At 10 wt% 2-MT, the viscous organic phase dominates the particle
morphology and AS remains dispersed at the surface edge, as shown in Fig. 4D. As organic wt%
increases to 45 wt%, morphology remains unchanged and the organic phase stays intact. The intact
core-shell morphology of 45 wt% 2-MT aerosol mimic contrasts with the well dispersed
morphology observed for 45 wt% 2-MTS aerosol mimic. For 2-MT, the organic diffusion is
limited under both sub- and supersaturated conditions, likely due to the undissolved viscous
organic phase (Fig. 4A). Specifically, 2-MT viscosity causes slower dissolution compared to AS
and results in the phase separated morphology. Thus, hygroscopicity of the 2-MT/AS mixture is
dominated by AS dissolution from the core and outer shell, corresponding to the hygroscopic
plateau observed for 2-MT/AS sub- and supersaturated water uptake measurements (Fig. 3).
Therefore, particle morphology and viscosity influence the synthesized 2-MT's ability to diffuse
through the aerosol droplet and can affect aerosol water uptake process. Indeed, a previous study
by Zhang et al. (2018) described the self-limiting effect of a core-shell morphology on IEPOX-
SOA reactive uptake and can now be observed in the 2-MT/AS water uptake process. However,
diffusion limitations can also result in the need for longer time periods to reach an equilibrium
state, as observed by dynamic surface tension measurements. Consequently, current
hygroscopicity measurements that occur at fast time scales may not capture the full water uptake
process of the synthesized organics and their mixtures. For example, the residence of aerosols
within DMT CCNC columns is ~ 10 s (Paramonov et al., 2015) while similar H-TDMA instrument
set ups have a residence time ~ 6.5 s (Mikhailov & Vlasenko, 2020). However, a previous study
by Chuang et al. (2003) found atmospheric droplet growth timescales range between 5 to 100 s,
congruent with the timescale of 2-MT and 2-MTS dynamic surface tension change (Fig. 2. and
Chuang, 2003). Therefore, hygroscopicity of viscous organic containing aerosols, such as 2-MT
and 2-MTS, must be studied at greater residence times to observe any possible effects on
hygroscopicity; understanding whether timescale effects CCN activity of organic-inorganic
aerosol mixtures can greatly impact current global models that may assume instantaneous solute
dissolution during the water uptake process. Furthermore, future studies should consider whether
the hygroscopicity approximations of viscous 2-MT/AS and 2-MTS/AS mixtures are time
dependent, as time-dependent droplet formation has been observed for biogenic aerosols (Vizenor
& Asa-Awuku, 2018). Currently, traditional $\kappa$-Köhler theory is unable to predict the water uptake
of 2-MT/AS and subsaturated 2-MTS/AS aerosols and does not consider solute and droplet kinetic
effects. However, by accounting for phase morphology and viscosity, $\kappa$ predictions may be
improved.
In addition, size-dependent morphology may also affect $\kappa$-hygroscopicity estimations. Several
studies observe a relationship between particle size and aerosol phase transitions during water
uptake (Veghte et al., 2013; Cheng et al., 2015; Altaf et al., 2016; Schmedding & Zuend, 2025).
Specifically, Veghte et al. (2013) and Cheng et al. (2015) observe smaller AS-organic particles
favoring a homogeneous liquid phase while larger particles remain in a partially engulfed
morphology; this finding correlates with 2-MT/AS engulfed morphology for particles imaged $\geq$
390 nm (Fig. 4). Indeed, for 2-MT/AS mixtures > 60 wt% 2-MT, $\kappa_{CCN}$ decreases with increasing
dry activation diameter before plateauing (Fig. S5). This trend may correlate to greater organic
diffusion as particle size and morphology changing before a dissolution limit is reached for > 60
wt% 2-MT/AS mixtures. For mixtures $\leq$ 60 wt% 2-MT, a similar decrease in $\kappa_{CCN}$ is observed
before hygroscopicity begins to increase; this may be attributed to the engulfed morphology in
larger particles (Fig. 4D-E) promoting AS dissolution and water uptake contribution while 2-MT
diffusion reaches a limit. However, the water uptake measurements performed in this study do not
account for size-dependent phase morphology in its analysis. Therefore, future work may build
upon the results of this study to better parameterize hygroscopicity based on initial particle size
and size-dependent phase morphology affecting $\kappa$-hygroscopicity estimations. In particular, size-
selected CCN measurements can be performed to better probe size-dependent morphology effects
on aerosol activation. By doing so, global models can incorporate these influential
physicochemical properties into predictions of aerosol-cloud interactions.
5. Summary and Implications
In this study, we investigated the influence of solute diffusivity and droplet phase morphology on
the hygroscopicity of synthesized 2-MT sample, 2-MTS sample, and their mixtures with AS.
Mixtures with AS were varied by organic wt%. Both 2-MT and 2-MTS were previously observed
to be viscous and glassy, affecting diffusivity through water. Additionally, previous studies found
2-MT to be weakly surface-active. To determine organic diffusivity and potential surface activity,
dynamic surface tension measurements were taken for aqueous organic and mixed organic-
inorganic solutions. 2-MT and 2-MTS were found to be weakly surface-active. Previous studies
by Bain et al., 2023 and Mikhailov et al., 2024 determined that surface activity in the dilute bulk
concentration range correlates with depressed aerosol surface tension. However, neither 2-MT
sample nor 2-MTS sample are sufficiently surface-active to depress droplet surface tension at the
air-surface interface. 2-MT and 2-MTS sample solutes move slowly in droplets and have estimated
diffusion rates ($D_s$) between $10^{-9}$ to $10^{-11}$ m$^2$ s$^{-1}$, with diffusion slowing as organic concentration is
increased. When mixed with AS, 2-MT diffusivity remains slow ($10^{-10}$ m$^2$ s$^{-1}$) while 2-MTS
diffusivity increases by an order of magnitude ($10^{-9}$ m$^2$ s$^{-1}$); 2-MTS diffusion in aqueous AS-
mixtures is similar to other quickly dissolving compounds, such as NaCl ($D_s = 10^{-9}$, Vitagliano &
Lyons, 1956; Leaist & Hao, 1992) and can result in a well-mixed droplet.
Organic viscosity and diffusion affect aerosol water uptake (Asa-Awuku & Nenes, 2007; Bones et
al., 2012; Tandon et al., 2019). For 2-MT, 2-MTS sample, and subsequent mixtures under both
sub- and supersaturated conditions, droplet growth is affected by solute diffusion. Subsaturated
droplet growth was measured using a H-TDMA at 88.2% RH and subsaturated hygroscopicity was
parameterized by $\kappa_{\text{H-TDMA}}$. For supersaturated conditions, a CCNC determined the activation ratio
of particles at varied supersaturations (0.3-1.4% SS) and water uptake was parameterized by $\kappa_{\text{CCN}}$.
2-MT/AS mixtures exhibit plateaued $\kappa_{\text{H-TDMA}}$ and $\kappa_{\text{CCN}}$ values close to $\kappa_{\text{int}}$ of AS (~0.61). A similar
plateau behavior is observed for 2-MTS/AS $\kappa_{\text{H-TDMA}}$. However, for supersaturated conditions, 2-
MTS/AS mixture $\kappa_{\text{CCN}}$ follows ideal mixing behavior, represented by its proximity to $\kappa$-
hygroscopicity predicted by $\kappa$-Köhler theory and volume additive ZSR. Additionally, $\kappa_{\text{H-TDMA}}$
remains higher than $\kappa_{\text{CCN}}$; this is a result of increased water content reducing viscosity effects and
enhancing organic dissolution under supersaturated conditions.
The $\kappa$-hygroscopicity plateau in Fig. 3 has been previously attributed to the presence of phase
separation, resulting in the inorganic, more soluble, and ideal compound (AS) driving water uptake
(Malek et al., 2023). However, for 2-MTS /AS ideal hygroscopic behavior is indicative of a well
dissolved, homogeneous droplet (Petters & Kreidenweis, 2007). To better understand phase
morphology of the synthesized organic-AS mixed particles, AFM measurements of synthesized 2-
MTS, 2-MTS/AS mixtures, and 2-MT/AS mixtures were acquired. 2-MTS aerosols are smooth,
spherical, viscous particles; when mixed with AS at 10 wt%, AS remains in the aqueous core and
is dispersed on the side of the particle, introducing roughness on the aerosol outer shell. As organic
concentration increases, the AS core is broken up through the particle. The less defined core-shell
morphology may be the result of AS disrupting the interactions between neighboring 2-MTS
particles in the viscous network; as a result, organic dissolution becomes faster as indicated by
greater 2-MTS diffusion rates. Thus, 2-MTS sample/AS aerosols behave similar to traditional full
dissolution assumptions. In comparison, 2-MT/AS mixture AFM images show an engulfed core-
shell morphology regardless of organic concentration. As a result, the viscous organic phase
remains intact while aqueous AS in the core drives hygroscopicity. A caveat to these results is the
presence of SMS, an organosulfate, being present within the 2-MTS sample at ~24 wt%. Therefore,
SMS may have an effect on surface tension, diffusivity, and hygroscopic trends observed for the
2-MTS sample/AS mixtures that is currently unknown in this study. Future work may utilize the
methodology laid out in this work to more deeply probe the influence of SMS and any additional
mixture component on viscous organic properties and water uptake.
This study demonstrates that viscosity can dictate organic diffusion through aqueous droplets,
resulting in complex phase morphology and water uptake properties. Furthermore, the synthesized
samples studied in this work a representative of the hygroscopic properties of IEPOX-SOA
mixtures. A recent study by Armstrong et al. (2025) determined that the IEPOX-SOA composition
is composed of a range of 2-MT, 2-MTS, and additional components that vary with aerosol acidity.
Thus, the synthesized samples present in this work may present a subset of SOA aerosols generated
and this study provides insight into its potential diffusive, hygroscopic, and phase behavior. For
example, as shown by this study's water uptake measurements, hygroscopicity from the
subsaturated to supersaturated regime evolves due to the presence of increased water content.
However, it is also noted that the hygroscopicity measurements performed in this study were on
short time scales (6-10 s); in comparison, dynamic surface tension measurements showed droplet
equilibrium being reached at 100-300 s for aqueous 2-MT, 2-MT/AS, and 2-MTS. Thus, current
water uptake measurements may not capture a potentially evolving hygroscopicity over time. This

is critical in understanding biogenic aerosol influence on cloud formation; a previous study by Chuang (2003) found that droplet formation can occur within time scales of 5-100 s, well within evolving diffusion times observed in this study. Therefore, future work must investigate potentially dynamic water uptake of viscous biogenic aerosols, such as 2-MT, 2-MTS. Furthermore, time dependent $\kappa$ can be developed to better account for organic diffusion within larger scale cloud parcel and global models. In addition to time dependency, $\kappa$-hygroscopicity estimations may also be affected by size dependent phase morphology. A study by Veghte et al. (2013) found smaller aerosol particles preferring a homogenous state, while larger particles have an engulfed core-shell morphology similar to 2-MT/AS aerosols in this study. Therefore, particle size may influence viscous organic-AS water uptake due to diffusion and morphological influences. Future work may explore and parameterize the effect of size-dependent phase separated morphology on aerosol activation through step size-selected CCN measurements. Ultimately, it is crucial to understand how biogenic aerosols, such as 2-MT and 2-MTS, properties (viscosity, diffusivity, and phase morphology) alter cloud formation. The results of this study demonstrate the co-dependency of these properties for two isoprene derived compounds and thus may improve our overall understanding of how biogenic aerosols, and their mixtures affect aerosol-cloud interactions.

**Author Contributions:** NF designed, collected, analyzed all experimental data, and analyzed theoretical models. CA contributed to design and analysis of water uptake data. SJ contributed to design and collection of H-TDMA experimental data. AD contributed to design and analysis of AFM data. MA contributed to design and collection of AFM data. ERR contributed to collection of surface tension experimental data. ZZ and AG contributed to sample synthesis. JLW contributed to design, collection, and analysis of dynamic surface tension experimental data. YZ contributed to viscosity, diffusion, and AFM data analysis. All authors contributed to the writing and preparation of the manuscript.

**Competing Interest Statement:** The authors have no competing interests to declare

**Acknowledgements:**

The authors acknowledge the support of this work by NSF under AGS #2131369, AGS #2124489, AGS #2131369 (Y.Z.), AGS #2131370 (J.S.), and AGS #2304669 (A.G., Z.Z., J.D.S.).

The authors acknowledge the characterization part of this work was performed in Texas A&M University Materials Characterization Core Facility (RRID:SCR_022202).

Abramson, E., Imre, D., Beránek, J., Wilson, J., & Zelenyuk, A. (2013). Experimental determination of chemical diffusion within secondary organic aerosol particles [10.1039/C2CP44013J].

*Physical Chemistry Chemical Physics*, *15*(8), 2983–2991. https://doi.org/10.1039/C2CP44013J

Albrecht, B. A. (1989). Aerosols, Cloud Microphysics, and Fractional Cloudiness. *Science*, *245*(4923), 1227–1230. https://doi.org/doi:10.1126/science.245.4923.1227

Altaf, M. B., Zuend, A., & Freedman, M. A. (2016). Role of nucleation mechanism on the size dependent morphology of organic aerosol [10.1039/C6CC03826C]. *Chemical Communications*, *52*(59), 9220–9223. https://doi.org/10.1039/C6CC03826C

Armstrong, N. C., Gagan, S., Dodero, A. J., Ferdousi-Rokib, N., Frauenheim, M., Gold, A., Zhang, Z., Asa-Awuku, A., Zhang, Y., & Surratt, J. D. (2025). Hygroscopicity Depends on Aerosol Acidity and Sulfate Content during the Reactive Uptake of Isoprene Epoxydiols. *ACS Earth and Space Chemistry*. https://doi.org/10.1021/acsearthspacechem.5c00163

Asa-Awuku, A., & Nenes, A. (2007). Effect of solute dissolution kinetics on cloud droplet formation: Extended Köhler theory. *Journal of Geophysical Research: Atmospheres*, *112*(D22). https://doi.org/https://doi.org/10.1029/2005JD006934

Bain, A., Ghosh, K., Prisle, N. L., & Bzdek, B. R. (2023). Surface-Area-to-Volume Ratio Determines Surface Tensions in Microscopic, Surfactant-Containing Droplets. *ACS Central Science*, *9*(11), 2076–2083. https://doi.org/10.1021/acscentsci.3c00998

Bain, A., Lalemi, L., Croll Dawes, N., Miles, R. E. H., Prophet, A. M., Wilson, K. R., & Bzdek, B. R. (2024). Surfactant Partitioning Dynamics in Freshly Generated Aerosol Droplets. *Journal of the American Chemical Society*, *146*(23), 16028–16038. https://doi.org/10.1021/jacs.4c03041

Beier, T., Cotter, E. R., Galloway, M. M., & Woo, J. L. (2019). In Situ Surface Tension Measurements of Hanging Droplet Methylglyoxal/Ammonium Sulfate Aerosol Mimics under Photooxidative Conditions. *ACS Earth and Space Chemistry*, *3*(7), 1208–1215. https://doi.org/10.1021/acsearthspacechem.9b00123

Bertram, A. K., Martin, S. T., Hanna, S. J., Smith, M. L., Bodsworth, A., Chen, Q., Kuwata, M., Liu, A., You, Y., & Zorn, S. R. (2011). Predicting the relative humidities of liquid-liquid phase separation, efflorescence, and deliquescence of mixed particles of ammonium sulfate, organic material, and water using the organic-to-sulfate mass ratio of the particle and the oxygen-to-carbon elemental ratio of the organic component. *Atmos. Chem. Phys.*, *11*(21), 10995–11006. https://doi.org/10.5194/acp-11-10995-2011

Bondy, A. L., Bonanno, D., Moffet, R. C., Wang, B., Laskin, A., & Ault, A. P. (2018). The diverse chemical mixing state of aerosol particles in the southeastern United States. *Atmos. Chem. Phys.*, *18*(16), 12595–12612. https://doi.org/10.5194/acp-18-12595-2018

Bones, D. L., Reid, J. P., Lienhard, D. M., & Krieger, U. K. (2012). Comparing the mechanism of water condensation and evaporation in glassy aerosol. *Proceedings of the National Academy of Sciences*, *109*(29), 11613–11618. https://doi.org/doi:10.1073/pnas.1200691109

Chan, M. N., Surratt, J. D., Claeys, M., Edgerton, E. S., Tanner, R. L., Shaw, S. L., Zheng, M., Knipping, E. M., Eddingsaas, N. C., Wennberg, P. O., & Seinfeld, J. H. (2010). Characterization and Quantification of Isoprene-Derived Epoxydiols in Ambient Aerosol in the Southeastern United States. *Environmental Science & Technology*, *44*(12), 4590–4596. https://doi.org/10.1021/es100596b

Chen, B., Mirrielees, J. A., Chen, Y., Onasch, T. B., Zhang, Z., Gold, A., Surratt, J. D., Zhang, Y., & Brooks, S. D. (2023). Glass Transition Temperatures of Organic Mixtures from Isoprene Epoxydiol-Derived Secondary Organic Aerosol. *The Journal of Physical Chemistry A*, *127*(18), 4125–4136. https://doi.org/10.1021/acs.jpca.2c08936

Chen, Y., Dombek, T., Hand, J., Zhang, Z., Gold, A., Ault, A. P., Levine, K. E., & Surratt, J. D. (2021). Seasonal Contribution of Isoprene-Derived Organosulfates to Total Water-Soluble Fine

Particulate Organic Sulfur in the United States. *ACS Earth and Space Chemistry*, *5*(9), 2419–2432. https://doi.org/10.1021/acsearthspacechem.1c00102

Cheng, Y., Su, H., Koop, T., Mikhailov, E., & Pöschl, U. (2015). Size dependence of phase transitions in aerosol nanoparticles. *Nature Communications*, *6*(1), 5923. https://doi.org/10.1038/ncomms6923

Chenyakin, Y., Ullmann, D. A., Evoy, E., Renbaum-Wolff, L., Kamal, S., & Bertram, A. K. (2017). Diffusion coefficients of organic molecules in sucrose–water solutions and comparison with Stokes–Einstein predictions. *Atmos. Chem. Phys.*, *17*(3), 2423–2435. https://doi.org/10.5194/acp-17-2423-2017

Chernyshev, V. S., & Skliar, M. (2015). Diffusivity Measurements of Solutes Impacting Interfacial Tension. *Industrial & Engineering Chemistry Research*, *54*(16), 4535–4544. https://doi.org/10.1021/ie504355w

Chuang, P. Y. (2003). Measurement of the timescale of hygroscopic growth for atmospheric aerosols. *Journal of Geophysical Research: Atmospheres*, *108*(D9). https://doi.org/https://doi.org/10.1029/2002JD002757

Claeys, M., Graham, B., Vas, G., Wang, W., Vermeylen, R., Pashynska, V., Cafmeyer, J., Guyon, P., Andreae, M. O., Artaxo, P., & Maenhaut, W. (2004). Formation of Secondary Organic Aerosols Through Photooxidation of Isoprene. *Science*, *303*(5661), 1173–1176. https://doi.org/doi:10.1126/science.1092805

Cooke, M. E., Armstrong, N. C., Lei, Z., Chen, Y., Waters, C. M., Zhang, Y., Buchenau, N. A., Dibley, M. Q., Ledsky, I. R., Szalkowski, T., Lee, J. Y., Baumann, K., Zhang, Z., Vizuete, W., Gold, A., Surratt, J. D., & Ault, A. P. (2022). Organosulfate Formation in Proxies for Aged Sea Spray Aerosol: Reactive Uptake of Isoprene Epoxydiols to Acidic Sodium Sulfate. *ACS Earth and Space Chemistry*, *6*(12), 2790–2800. https://doi.org/10.1021/acsearthspacechem.2c00156

Curry, L. A., Tsui, W. G., & McNeill, V. F. (2018). Technical note: Updated parameterization of the reactive uptake of glyoxal and methylglyoxal by atmospheric aerosols and cloud droplets. *Atmos. Chem. Phys.*, *18*(13), 9823–9830. https://doi.org/10.5194/acp-18-9823-2018

Cziczo, D. J., Nowak, J. B., Hu, J. H., & Abbatt, J. P. D. (1997). Infrared spectroscopy of model tropospheric aerosols as a function of relative humidity: Observation of deliquescence and crystallization. *Journal of Geophysical Research: Atmospheres*, *102*(D15), 18843–18850. https://doi.org/https://doi.org/10.1029/97JD01361

DeCarlo, P. F., Slowik, J. G., Worsnop, D. R., Davidovits, P., & Jimenez, J. L. (2004). Particle Morphology and Density Characterization by Combined Mobility and Aerodynamic Diameter Measurements. Part 1: Theory. *Aerosol Science and Technology*, *38*(12), 1185–1205. https://doi.org/10.1080/027868290903907

DeRieux, W. S. W., Li, Y., Lin, P., Laskin, J., Laskin, A., Bertram, A. K., Nizkorodov, S. A., & Shiraiwa, M. (2018). Predicting the glass transition temperature and viscosity of secondary organic material using molecular composition. *Atmos. Chem. Phys.*, *18*(9), 6331–6351. https://doi.org/10.5194/acp-18-6331-2018

Eastoe, J., Dalton, J. S., Rogueda, P. G. A., & Griffiths, P. C. (1998). Evidence for Activation−Diffusion Controlled Dynamic Surface Tension with a Nonionic Surfactant. *Langmuir*, *14*(5), 979–981. https://doi.org/10.1021/la971241w

Einstein, A. (1905). Über die von der molekularkinetischen Theorie der Wärme geforderte Bewegung von in ruhenden Flüssigkeiten suspendierten Teilchen. *Annalen der Physik*, *322*(8), 549–560. https://doi.org/https://doi.org/10.1002/andp.19053220806

El Haber, M., Ferronato, C., Giroir-Fendler, A., Fine, L., & Nozière, B. (2023). Salting out, non-ideality and synergism enhance surfactant efficiency in atmospheric aerosols. *Scientific Reports*, *13*(1), 20672. https://doi.org/10.1038/s41598-023-48040-5

Ferdousi-Rokib, N., A. Malek, K., Mitchell, I., M. Fierce, L., & Asa-Awuku, A. A. (2025). Aerosol
Hygroscopicity and Surface-Active Coverage for the Droplet Growth of Aerosol Mixtures. *ACS*
*ES&T Air*, *2*(8), 1454–1467. https://doi.org/10.1021/acsestair.4c00303
Fertil, D., Pierre-Louis, K., Ingwer, S., Galloway, M. M., & Woo, J. L. (2025). Surfactant Effects in
Irradiated, Hanging-Droplet, Aqueous-Phase Glyoxal/Ammonium Sulfate Aerosol Mimic
Systems. *ACS Earth and Space Chemistry*.
https://doi.org/10.1021/acsearthspacechem.4c00288
Fordham, S., & Freeth, F. A. (1948). On the calculation of surface tension from measurements of
pendant drops. *Proceedings of the Royal Society of London. Series A. Mathematical and*
*Physical Sciences*, *194*(1036), 1–16. https://doi.org/doi:10.1098/rspa.1948.0063
Freedman, M. A. (2017). Phase separation in organic aerosol [10.1039/C6CS00783J]. *Chemical*
*Society Reviews*, *46*(24), 7694–7705. https://doi.org/10.1039/C6CS00783J
Froyd, K. D., Murphy, S. M., Murphy, D. M., de Gouw, J. A., Eddingsaas, N. C., & Wennberg, P. O.
(2010). Contribution of isoprene-derived organosulfates to free tropospheric aerosol mass.
*Proceedings of the National Academy of Sciences*, *107*(50), 21360–21365.
https://doi.org/doi:10.1073/pnas.1012561107
Fuchs, N. A. (1963). On the stationary charge distribution on aerosol particles in a bipolar ionic
atmosphere. *Geofisica pura e applicata*, *56*(1), 185–193.
https://doi.org/10.1007/BF01993343
Gohil, K. (2022). *kgohil27/PyCAT: v1.0 (v1.0)*. Zenodo.
Gohil, K., & Asa-Awuku, A. A. (2022). Cloud condensation nuclei (CCN) activity analysis of low-
hygroscopicity aerosols using the aerodynamic aerosol classifier (AAC). *Atmos. Meas. Tech.*,
*15*(4), 1007–1019. https://doi.org/10.5194/amt-15-1007-2022
Guenther, A. B., Jiang, X., Heald, C. L., Sakulyanontvittaya, T., Duhl, T., Emmons, L. K., & Wang, X.
(2012). The Model of Emissions of Gases and Aerosols from Nature version 2.1 (MEGAN2.1):
an extended and updated framework for modeling biogenic emissions. *Geosci. Model Dev.*,
*5*(6), 1471–1492. https://doi.org/10.5194/gmd-5-1471-2012
Hettiyadura, A. P. S., Al-Naiema, I. M., Hughes, D. D., Fang, T., & Stone, E. A. (2019). Organosulfates
in Atlanta, Georgia: anthropogenic influences on biogenic secondary organic aerosol
formation. *Atmos. Chem. Phys.*, *19*(5), 3191–3206. https://doi.org/10.5194/acp-19-3191-
862     2019

Hughes, D. D., Christiansen, M. B., Milani, A., Vermeuel, M. P., Novak, G. A., Alwe, H. D., Dickens, A.
F., Pierce, R. B., Millet, D. B., Bertram, T. H., Stanier, C. O., & Stone, E. A. (2021). PM2.5
chemistry, organosulfates, and secondary organic aerosol during the 2017 Lake Michigan
Ozone Study. *Atmospheric Environment*, *244*, 117939.
https://doi.org/https://doi.org/10.1016/j.atmosenv.2020.117939
Hyvärinen, A.-P., Raatikainen, T., Laaksonen, A., Viisanen, Y., & Lihavainen, H. (2005). Surface
tensions and densities of H2SO4 + NH3 + water solutions. *Geophysical Research Letters*,
*32*(16). https://doi.org/https://doi.org/10.1029/2005GL023268
Intergovernmental Panel on Climate, C. (2023). *Climate Change 2021 – The Physical Science Basis:*
*Working Group I Contribution to the Sixth Assessment Report of the Intergovernmental Panel*
*on Climate Change*. Cambridge University Press. https://doi.org/DOI:
874     10.1017/9781009157896

Jeong, R., Lilek, J., Zuend, A., Xu, R., Chan, M. N., Kim, D., Moon, H. G., & Song, M. (2022). Viscosity
and physical state of sucrose mixed with ammonium sulfate droplets. *Atmos. Chem. Phys.*,
*22*(13), 8805–8817. https://doi.org/10.5194/acp-22-8805-2022
Joos, P., & Rillaerts, E. (1981). Theory on the determination of the dynamic surface tension with the
drop volume and maximum bubble pressure methods. *Journal of Colloid and Interface*
*Science*, *79*(1), 96–100. https://doi.org/https://doi.org/10.1016/0021-9797(81)90051-5
Kampf, C. J., Waxman, E. M., Slowik, J. G., Dommen, J., Pfaffenberger, L., Praplan, A. P., Prévôt, A. S.
H., Baltensperger, U., Hoffmann, T., & Volkamer, R. (2013). Effective Henry's Law Partitioning
and the Salting Constant of Glyoxal in Aerosols Containing Sulfate. *Environmental Science &*
*Technology*, *47*(9), 4236–4244. https://doi.org/10.1021/es400083d
Kanakidou, M., Seinfeld, J. H., Pandis, S. N., Barnes, I., Dentener, F. J., Facchini, M. C., Van Dingenen,
R., Ervens, B., Nenes, A., Nielsen, C. J., Swietlicki, E., Putaud, J. P., Balkanski, Y., Fuzzi, S.,
Horth, J., Moortgat, G. K., Winterhalter, R., Myhre, C. E. L., Tsigaridis, K., . . . Wilson, J. (2005).
Organic aerosol and global climate modelling: a review. *Atmos. Chem. Phys.*, *5*(4), 1053–
1123. https://doi.org/10.5194/acp-5-1053-2005
Kang, B., Tang, H., Zhao, Z., & Song, S. (2020). Hofmeister Series: Insights of Ion Specificity from
Amphiphilic Assembly and Interface Property. *ACS Omega*, *5*(12), 6229–6239.
https://doi.org/10.1021/acsomega.0c00237
Kleinheins, J., Shardt, N., Lohmann, U., & Marcolli, C. (2025). The surface tension and cloud
condensation nuclei (CCN) activation of sea spray aerosol particles. *Atmos. Chem. Phys.*,
*25*(2), 881–903. https://doi.org/10.5194/acp-25-881-2025
Köhler, H. (1936). The nucleus in and the growth of hygroscopic droplets [10.1039/TF9363201152].
*Transactions of the Faraday Society*, *32*(0), 1152–1161.
https://doi.org/10.1039/TF9363201152
Kreidenweis, S. M., & Asa-Awuku, A. (2014). 5.13 - Aerosol Hygroscopicity: Particle Water Content
and Its Role in Atmospheric Processes. In H. D. Holland & K. K. Turekian (Eds.), *Treatise on*
*Geochemistry (Second Edition)* (pp. 331–361). Elsevier.
https://doi.org/https://doi.org/10.1016/B978-0-08-095975-7.00418-6
Lance, S., Nenes, A., Medina, J., & Smith, J. N. (2006). Mapping the Operation of the DMT Continuous
Flow CCN Counter. *Aerosol Science and Technology*, *40*(4), 242–254.
https://doi.org/10.1080/02786820500543290
Laskina, O., Morris, H. S., Grandquist, J. R., Qin, Z., Stone, E. A., Tivanski, A. V., & Grassian, V. H.
(2015). Size Matters in the Water Uptake and Hygroscopic Growth of Atmospherically
Relevant Multicomponent Aerosol Particles. *The Journal of Physical Chemistry A*, *119*(19),
4489–4497. https://doi.org/10.1021/jp510268p
Leaist, D. G., & Hao, L. (1992). Binary mutual diffusion coefficients of aqueous ammonium and
potassium sulfates at 25°C. *Journal of Solution Chemistry*, *21*(4), 345–350.
https://doi.org/10.1007/BF00647857
Malek, K., Gohil, K., Olonimoyo, E. A., Ferdousi-Rokib, N., Huang, Q., Pitta, K. R., Nandy, L., Voss, K.
A., Raymond, T. M., Dutcher, D. D., Freedman, M. A., & Asa-Awuku, A. (2023). Liquid–Liquid
Phase Separation Can Drive Aerosol Droplet Growth in Supersaturated Regimes. *ACS*
*Environmental Au*. https://doi.org/10.1021/acsenvironau.3c00015
Mikhailov, E. F., & Vlasenko, S. S. (2020). High-humidity tandem differential mobility analyzer for
accurate determination of aerosol hygroscopic growth, microstructure, and activity
coefficients over a wide range of relative humidity. *Atmos. Meas. Tech.*, *13*(4), 2035–2056.
https://doi.org/10.5194/amt-13-2035-2020
Mikhailov, E. F., Vlasenko, S. S., & Kiselev, A. A. (2024). Water activity and surface tension of aqueous
ammonium sulfate and D-glucose aerosol nanoparticles. *Atmos. Chem. Phys.*, *24*(5), 2971–
2984. https://doi.org/10.5194/acp-24-2971-2024

Moore, R. H., Nenes, A., & Medina, J. (2010). Scanning Mobility CCN Analysis—A Method for Fast Measurements of Size-Resolved CCN Distributions and Activation Kinetics. *Aerosol Science and Technology*, *44*(10), 861–871. https://doi.org/10.1080/02786826.2010.498715

Murphy, D. M., Thomson, D. S., & Mahoney, M. J. (1998). In Situ Measurements of Organics, Meteoritic Material, Mercury, and Other Elements in Aerosols at 5 to 19 Kilometers. *Science*, *282*(5394), 1664–1669. https://doi.org/doi:10.1126/science.282.5394.1664

O'Meara, S., Topping, D. O., & McFiggans, G. (2016). The rate of equilibration of viscous aerosol particles. *Atmos. Chem. Phys.*, *16*(8), 5299–5313. https://doi.org/10.5194/acp-16-5299-2016

Ott, E.-J. E., Tackman, E. C., & Freedman, M. A. (2020). Effects of Sucrose on Phase Transitions of Organic/Inorganic Aerosols. *ACS Earth and Space Chemistry*, *4*(4), 591–601. https://doi.org/10.1021/acsearthspacechem.0c00006

Padró, L. T., Tkacik, D., Lathem, T., Hennigan, C. J., Sullivan, A. P., Weber, R. J., Huey, L. G., & Nenes, A. (2010). Investigation of cloud condensation nuclei properties and droplet growth kinetics of the water-soluble aerosol fraction in Mexico City. *Journal of Geophysical Research: Atmospheres*, *115*(D9). https://doi.org/https://doi.org/10.1029/2009JD013195

Paramonov, M., Kerminen, V. M., Gysel, M., Aalto, P. P., Andreae, M. O., Asmi, E., Baltensperger, U., Bougiatioti, A., Brus, D., Frank, G. P., Good, N., Gunthe, S. S., Hao, L., Irwin, M., Jaatinen, A., Jurányi, Z., King, S. M., Kortelainen, A., Kristensson, A., . . . Sierau, B. (2015). A synthesis of cloud condensation nuclei counter (CCNC) measurements within the EUCAARI network. *Atmos. Chem. Phys.*, *15*(21), 12211–12229. https://doi.org/10.5194/acp-15-12211-2015

Paulot, F., Crounse, J. D., Kjaergaard, H. G., Kürten, A., St. Clair, J. M., Seinfeld, J. H., & Wennberg, P. O. (2009). Unexpected Epoxide Formation in the Gas-Phase Photooxidation of Isoprene. *Science*, *325*(5941), 730–733. https://doi.org/doi:10.1126/science.1172910

Peng, C., Razafindrambinina, P. N., Malek, K. A., Chen, L., Wang, W., Huang, R. J., Zhang, Y., Ding, X., Ge, M., Wang, X., Asa-Awuku, A. A., & Tang, M. (2021). Interactions of organosulfates with water vapor under sub- and supersaturated conditions. *Atmos. Chem. Phys.*, *21*(9), 7135–7148. https://doi.org/10.5194/acp-21-7135-2021

Petters, M. D., & Kreidenweis, S. M. (2007). A single parameter representation of hygroscopic growth and cloud condensation nucleus activity. *Atmos. Chem. Phys.*, *7*(8), 1961–1971. https://doi.org/10.5194/acp-7-1961-2007

Pratt, K. A., & Prather, K. A. (2010). Aircraft measurements of vertical profiles of aerosol mixing states. *Journal of Geophysical Research: Atmospheres*, *115*(D11). https://doi.org/https://doi.org/10.1029/2009JD013150

Prisle, N., & Mølgaard, B. (2018). Modeling CCN activity of chemically unresolved model HULIS, including surface tension, non-ideality, and surface partitioning. *Atmospheric Chemistry and Physics Discussions*, 1–23. https://doi.org/10.5194/acp-2018-789

Prisle, N. L., Engelhart, G. J., Bilde, M., & Donahue, N. M. (2010). Humidity influence on gas-particle phase partitioning of α-pinene + O3 secondary organic aerosol. *Geophysical Research Letters*, *37*(1). https://doi.org/https://doi.org/10.1029/2009GL041402

Pruppacher, H. R., Klett, J. D., & Springer. (1997). *Microphysics of Clouds and Precipitation*. Springer. https://books.google.com/books?id=Nk40jwEACAAJ

Reid, J. P., Bertram, A. K., Topping, D. O., Laskin, A., Martin, S. T., Petters, M. D., Pope, F. D., & Rovelli, G. (2018). The viscosity of atmospherically relevant organic particles. *Nature Communications*, *9*(1), 956. https://doi.org/10.1038/s41467-018-03027-z

Renbaum-Wolff, L., Grayson, J. W., Bateman, A. P., Kuwata, M., Sellier, M., Murray, B. J., Shilling, J. E., Martin, S. T., & Bertram, A. K. (2013). Viscosity of α-pinene secondary organic material and

implications for particle growth and reactivity. *Proceedings of the National Academy of Sciences*, *110*(20), 8014–8019. https://doi.org/doi:10.1073/pnas.1219548110

Riemer, N., Ault, A. P., West, M., Craig, R. L., & Curtis, J. H. (2019). Aerosol Mixing State: Measurements, Modeling, and Impacts. *Reviews of Geophysics*, *57*(2), 187–249. https://doi.org/https://doi.org/10.1029/2018RG000615

Riipinen, I., Pierce, J. R., Yli-Juuti, T., Nieminen, T., Häkkinen, S., Ehn, M., Junninen, H., Lehtipalo, K., Petäjä, T., Slowik, J., Chang, R., Shantz, N. C., Abbatt, J., Leaitch, W. R., Kerminen, V. M., Worsnop, D. R., Pandis, S. N., Donahue, N. M., & Kulmala, M. (2011). Organic condensation: a vital link connecting aerosol formation to cloud condensation nuclei (CCN) concentrations. *Atmos. Chem. Phys.*, *11*(8), 3865–3878. https://doi.org/10.5194/acp-11-3865-2011

Riva, M., Chen, Y., Zhang, Y., Lei, Z., Olson, N. E., Boyer, H. C., Narayan, S., Yee, L. D., Green, H. S., Cui, T., Zhang, Z., Baumann, K., Fort, M., Edgerton, E., Budisulistiorini, S. H., Rose, C. A., Ribeiro, I. O., e Oliveira, R. L., dos Santos, E. O., . . . Surratt, J. D. (2019). Increasing Isoprene Epoxydiol-to-Inorganic Sulfate Aerosol Ratio Results in Extensive Conversion of Inorganic Sulfate to Organosulfur Forms: Implications for Aerosol Physicochemical Properties. *Environmental Science & Technology*, *53*(15), 8682–8694. https://doi.org/10.1021/acs.est.9b01019

Roberts, G. C., & Nenes, A. (2005). A Continuous-Flow Streamwise Thermal-Gradient CCN Chamber for Atmospheric Measurements. *Aerosol Science and Technology*, *39*(3), 206–221. https://doi.org/10.1080/027868290913988

Rose, D., Gunthe, S. S., Mikhailov, E., Frank, G. P., Dusek, U., Andreae, M. O., & Pöschl, U. (2008). Calibration and measurement uncertainties of a continuous-flow cloud condensation nuclei counter (DMT-CCNC): CCN activation of ammonium sulfate and sodium chloride aerosol particles in theory and experiment. *Atmos. Chem. Phys.*, *8*(5), 1153–1179. https://doi.org/10.5194/acp-8-1153-2008

Ross, S. (1945). The Change of Surface Tension with Time. I. Theories of Diffusion to the Surface. *Journal of the American Chemical Society*, *67*(6), 990–994. https://doi.org/10.1021/ja01222a031

Ruehl, C. R., Chuang, P. Y., Nenes, A., Cappa, C. D., Kolesar, K. R., & Goldstein, A. H. (2012). Strong evidence of surface tension reduction in microscopic aqueous droplets. *Geophysical Research Letters*, *39*(23). https://doi.org/https://doi.org/10.1029/2012GL053706

Ruehl, C. R., Davies, J. F., & Wilson, K. R. (2016). An interfacial mechanism for cloud droplet formation on organic aerosols. *Science*, *351*(6280), 1447–1450. https://doi.org/doi:10.1126/science.aad4889

Saukko, E., Lambe, A. T., Massoli, P., Koop, T., Wright, J. P., Croasdale, D. R., Pedernera, D. A., Onasch, T. B., Laaksonen, A., Davidovits, P., Worsnop, D. R., & Virtanen, A. (2012). Humidity-dependent phase state of SOA particles from biogenic and anthropogenic precursors. *Atmos. Chem. Phys.*, *12*(16), 7517–7529. https://doi.org/10.5194/acp-12-7517-2012

Saxena, P., et al. (1995). Organics alter hygroscopic behavior of atmospheric particles. *Journal of Geophysical Research: Atmospheres*, *100*(D9), 18755–18770. https://doi.org/https://doi.org/10.1029/95JD01835

Schmedding, R., & Zuend, A. (2025). The role of interfacial tension in the size-dependent phase separation of atmospheric aerosol particles. *Atmos. Chem. Phys.*, *25*(1), 327–346. https://doi.org/10.5194/acp-25-327-2025

Seinfeld, J., & Pandis, S. (1998). *Atmospheric Chemistry and Physics: From Air Pollution to Climate Change*.

Seinfeld, J. H. (2003). TROPOSPHERIC CHEMISTRY AND COMPOSITION | Aerosols/Particles. In J. R.
Holton (Ed.), *Encyclopedia of Atmospheric Sciences* (pp. 2349–2354). Academic Press.
https://doi.org/https://doi.org/10.1016/B0-12-227090-8/00438-3
Sheldon, C. S., Choczynski, J. M., Morton, K., Palacios Diaz, T., Davis, R. D., & Davies, J. F. (2023).
Exploring the hygroscopicity, water diffusivity, and viscosity of organic–inorganic aerosols –
a case study on internally-mixed citric acid and ammonium sulfate particles
[10.1039/D2EA00116K]. *Environmental Science: Atmospheres*, *3*(1), 24–34.
https://doi.org/10.1039/D2EA00116K
Shiraiwa, M., Li, Y., Tsimpidi, A. P., Karydis, V. A., Berkemeier, T., Pandis, S. N., Lelieveld, J., Koop, T.,
& Pöschl, U. (2017). Global distribution of particle phase state in atmospheric secondary
organic aerosols. *Nature Communications*, *8*(1), 15002.
https://doi.org/10.1038/ncomms15002
Shiraiwa, M., & Seinfeld, J. H. (2012). Equilibration timescale of atmospheric secondary organic
aerosol partitioning. *Geophysical Research Letters*, *39*(24).
https://doi.org/https://doi.org/10.1029/2012GL054008
Sindelarova, K., Granier, C., Bouarar, I., Guenther, A., Tilmes, S., Stavrakou, T., Müller, J. F., Kuhn,
U., Stefani, P., & Knorr, W. (2014). Global data set of biogenic VOC emissions calculated by
the MEGAN model over the last 30 years. *Atmos. Chem. Phys.*, *14*(17), 9317–9341.
https://doi.org/10.5194/acp-14-9317-2014
Song, M., Marcolli, C., Krieger, U. K., Lienhard, D. M., & Peter, T. (2013). Morphologies of mixed
organic/inorganic/aqueous aerosol droplets [10.1039/C3FD00049D]. *Faraday Discussions*,
*165*(0), 289–316. https://doi.org/10.1039/C3FD00049D
Spelt, J. (1996). *Applied Surface Thermodynamics*. Crc Press.
Srivastava, D., Vu, T. V., Tong, S., Shi, Z., & Harrison, R. M. (2022). Formation of secondary organic
aerosols from anthropogenic precursors in laboratory studies. *npj Climate and Atmospheric
Science*, *5*(1), 22. https://doi.org/10.1038/s41612-022-00238-6
Sullivan, R. C., Moore, M. J. K., Petters, M. D., Kreidenweis, S. M., Roberts, G. C., & Prather, K. A.
(2009). Effect of chemical mixing state on the hygroscopicity and cloud nucleation properties
of calcium mineral dust particles. *Atmos. Chem. Phys.*, *9*(10), 3303–3316.
https://doi.org/10.5194/acp-9-3303-2009
Tandon, A., Rothfuss, N. E., & Petters, M. D. (2019). The effect of hydrophobic glassy organic material
on the cloud condensation nuclei activity of particles with different morphologies. *Atmos.
Chem. Phys.*, *19*(5), 3325–3339. https://doi.org/10.5194/acp-19-3325-2019
Toivola, M., Prisle, N. L., Elm, J., Waxman, E. M., Volkamer, R., & Kurtén, T. (2017). Can COSMOTherm
Predict a Salting in Effect? *The Journal of Physical Chemistry A, 121*(33), 6288–6295.
https://doi.org/10.1021/acs.jpca.7b04847
Topping, D. (2010). An analytical solution to calculate bulk mole fractions for any number of
components in aerosol droplets after considering partitioning to a surface layer. *Geosci.
Model Dev.*, *3*(2), 635–642. https://doi.org/10.5194/gmd-3-635-2010
Topping, D. O., McFiggans, G. B., Kiss, G., Varga, Z., Facchini, M. C., Decesari, S., & Mircea, M. (2007).
Surface tensions of multi-component mixed inorganic/organic aqueous systems of
atmospheric significance: measurements, model predictions and importance for cloud
activation predictions. *Atmos. Chem. Phys.*, *7*(9), 2371–2398. https://doi.org/10.5194/acp-
1061    7-2371-2007

Twomey, S. (1959). The nuclei of natural cloud formation part II: The supersaturation in natural
clouds and the variation of cloud droplet concentration. *Geofisica pura e applicata*, *43*(1),
243–249. https://doi.org/10.1007/BF01993560

Twomey, S. (1974). Pollution and the planetary albedo. *Atmospheric Environment (1967)*, *8*(12), 1251–1256. https://doi.org/https://doi.org/10.1016/0004-6981(74)90004-3

Veghte, D. P., Altaf, M. B., & Freedman, M. A. (2013). Size Dependence of the Structure of Organic Aerosol. *Journal of the American Chemical Society*, *135*(43), 16046–16049. https://doi.org/10.1021/ja408903g

Vepsäläinen, S., Calderón, S. M., & Prisle, N. L. (2023). Comparison of six approaches to predicting droplet activation of surface active aerosol – Part 2: Strong surfactants. *Atmos. Chem. Phys.*, *23*(23), 15149–15164. https://doi.org/10.5194/acp-23-15149-2023

Vignes, A. (1966). Diffusion in Binary Solutions. Variation of Diffusion Coefficient with Composition. *Industrial & Engineering Chemistry Fundamentals*, *5*(2), 189–199. https://doi.org/10.1021/i160018a007

Virtanen, A., Joutsensaari, J., Koop, T., Kannosto, J., Yli-Pirilä, P., Leskinen, J., Mäkelä, J. M., Holopainen, J. K., Pöschl, U., Kulmala, M., Worsnop, D. R., & Laaksonen, A. (2010). An amorphous solid state of biogenic secondary organic aerosol particles. *Nature*, *467*(7317), 824–827. https://doi.org/10.1038/nature09455

Vitagliano, V., & Lyons, P. A. (1956). Diffusion Coefficients for Aqueous Solutions of Sodium Chloride and Barium Chloride. *Journal of the American Chemical Society*, *78*(8), 1549–1552. https://doi.org/10.1021/ja01589a011

Vizenor, A. E., & Asa-Awuku, A. A. (2018). Gas-phase kinetics modifies the CCN activity of a biogenic SOA [10.1039/C8CP00075A]. *Physical Chemistry Chemical Physics*, *20*(9), 6591–6597. https://doi.org/10.1039/C8CP00075A

Wallace, B. J., Price, C. L., Davies, J. F., & Preston, T. C. (2021). Multicomponent diffusion in atmospheric aerosol particles [10.1039/D0EA00008F]. *Environmental Science: Atmospheres*, *1*(1), 45–55. https://doi.org/10.1039/D0EA00008F

Wang, M., Wu, P., Sengupta, S. S., Chadhary, B. I., Cogen, J. M., & Li, B. (2011). Investigation of Water Diffusion in Low-Density Polyethylene by Attenuated Total Reflectance Fourier Transform Infrared Spectroscopy and Two-Dimensional Correlation Analysis. *Industrial & Engineering Chemistry Research*, *50*(10), 6447–6454. https://doi.org/10.1021/ie102221a

Werner, E. K., Hammond, M., & Bain, A. (2025). Surface tension predictions during hygroscopic growth and cloud droplet activation using a simple kinetic surfactant partitioning model. *Aerosol Science and Technology*, *59*(7), 781–793. https://doi.org/10.1080/02786826.2025.2465705

Wex, H., Stratmann, F., Topping, D., & McFiggans, G. (2008). The Kelvin versus the Raoult Term in the Köhler Equation. *Journal of the Atmospheric Sciences*, *65*(12), 4004–4016. https://doi.org/https://doi.org/10.1175/2008JAS2720.1

Wiedensohler, A. (1988). An approximation of the bipolar charge distribution for particles in the submicron size range. *Journal of Aerosol Science*, *19*, 387–389.

Wolf, M. J., Zhang, Y., Zhou, J., Surratt, J. D., Turpin, B. J., & Cziczo, D. J. (2021). Enhanced Ice Nucleation of Simulated Sea Salt Particles with the Addition of Anthropogenic Per- and Polyfluoroalkyl Substances. *ACS Earth and Space Chemistry*, *5*(8), 2074–2085. https://doi.org/10.1021/acsearthspacechem.1c00138

Wu, L., Li, X., Kim, H., Geng, H., Godoi, R. H. M., Barbosa, C. G. G., Godoi, A. F. L., Yamamoto, C. I., de Souza, R. A. F., Pöhlker, C., Andreae, M. O., & Ro, C. U. (2019). Single-particle characterization of aerosols collected at a remote site in the Amazonian rainforest and an urban site in Manaus, Brazil. *Atmos. Chem. Phys.*, *19*(2), 1221–1240. https://doi.org/10.5194/acp-19-1221-2019

Yang, F., Chen, H., Wang, X., Yang, X., Du, J., & Chen, J. (2009). Single particle mass spectrometry of oxalic acid in ambient aerosols in Shanghai: Mixing state and formation mechanism.

*Atmospheric Environment*, *43*(25), 3876–3882.
https://doi.org/https://doi.org/10.1016/j.atmosenv.2009.05.002
Zhang, C., Lu, M., Ma, N., Yang, Y., Wang, Y., Größ, J., Fan, Z., Wang, M., & Wiedensohler, A. (2023).
Hygroscopicity of aerosol particles composed of surfactant SDS and its internal mixture with
ammonium sulfate at relative humidities up to 99.9%. *Atmospheric Environment*, *298*,
119625. https://doi.org/https://doi.org/10.1016/j.atmosenv.2023.119625
Zhang, Q., Jimenez, J. L., Canagaratna, M. R., Allan, J. D., Coe, H., Ulbrich, I., Alfarra, M. R., Takami,
A., Middlebrook, A. M., Sun, Y. L., Dzepina, K., Dunlea, E., Docherty, K., DeCarlo, P. F.,
Salcedo, D., Onasch, T., Jayne, J. T., Miyoshi, T., Shimono, A., . . . Worsnop, D. R. (2007).
Ubiquity and dominance of oxygenated species in organic aerosols in anthropogenically-
influenced Northern Hemisphere midlatitudes. *Geophysical Research Letters*, *34*(13).
https://doi.org/https://doi.org/10.1029/2007GL029979
Zhang, Y., Chen, Y., Lambe, A. T., Olson, N. E., Lei, Z., Craig, R. L., Zhang, Z., Gold, A., Onasch, T. B.,
Jayne, J. T., Worsnop, D. R., Gaston, C. J., Thornton, J. A., Vizuete, W., Ault, A. P., & Surratt, J.
D. (2018). Effect of the Aerosol-Phase State on Secondary Organic Aerosol Formation from
the Reactive Uptake of Isoprene-Derived Epoxydiols (IEPOX). *Environmental Science &*
*Technology Letters*, *5*(3), 167–174. https://doi.org/10.1021/acs.estlett.8b00044
Zhang, Y., Chen, Y., Lei, Z., Olson, N. E., Riva, M., Koss, A. R., Zhang, Z., Gold, A., Jayne, J. T.,
Worsnop, D. R., Onasch, T. B., Kroll, J. H., Turpin, B. J., Ault, A. P., & Surratt, J. D. (2019a).
Joint Impacts of Acidity and Viscosity on the Formation of Secondary Organic Aerosol from
Isoprene Epoxydiols (IEPOX) in Phase Separated Particles. *ACS Earth and Space Chemistry*,
*3*(12), 2646–2658. https://doi.org/10.1021/acsearthspacechem.9b00209
Zhang, Y., Nichman, L., Spencer, P., Jung, J. I., Lee, A., Heffernan, B. K., Gold, A., Zhang, Z., Chen, Y.,
Canagaratna, M. R., Jayne, J. T., Worsnop, D. R., Onasch, T. B., Surratt, J. D., Chandler, D.,
Davidovits, P., & Kolb, C. E. (2019b). The Cooling Rate- and Volatility-Dependent Glass-
Forming Properties of Organic Aerosols Measured by Broadband Dielectric Spectroscopy.
*Environmental Science & Technology*, *53*(21), 12366–12378.
https://doi.org/10.1021/acs.est.9b03317
Zhang, Y., Sanchez, M. S., Douet, C., Wang, Y., Bateman, A. P., Gong, Z., Kuwata, M., Renbaum-
Wolff, L., Sato, B. B., Liu, P. F., Bertram, A. K., Geiger, F. M., & Martin, S. T. (2015). Changing
shapes and implied viscosities of suspended submicron particles. *Atmos. Chem. Phys.*,
*15*(14), 7819–7829. https://doi.org/10.5194/acp-15-7819-2015
Zhu, J., Penner, J. E., Lin, G., Zhou, C., Xu, L., & Zhuang, B. (2017). Mechanism of SOA formation
determines magnitude of radiative effects. *Proceedings of the National Academy of*
*Sciences*, *114*(48), 12685–12690. https://doi.org/doi:10.1073/pnas.1712273114