# Peer review of "Main Manuscript for 1 Hygroscopicity of Isoprene-Derived Secondary Organic Aerosol Mixture Proxies: 2 Importance of Diffusion and Salting In Effects 3 Nahin Ferdousi-Rokib1\*, N. Cazimir Armstrong2, Stephanie Jacoby3, Alana J. Dodero4, Martin 4 Ahn<sup"

_EGUsphere, 2025_

## Author Comment (AC1)

Reviewer 1

We thank the reviewer for their positive comments and questions. We have addressed each comment on a point by point basis in blue text.

In this study, the authors demonstrates that viscosity could influence the diffusion within aqueous droplets, resulting in complex phase morphology and water uptake properties for the 2-MT, 2-MTS and their mixtures with AS under saturated and supersaturated conditions. This is a comprehensive laboratory work and provides new valuable data to better understand the water uptake of isoprene derived SOA. The paper is well written. The data are well presented and discussed. I have some questions related to the measurements and a few minor comments.

Comments on measurements

Line 150, "Droplet surface tension ($\sigma_{s/a}$) was measured using a pendant drop tensiometer with a modified profile analysis tensiometer (SINTERFACE Inc.); the experimental set up has been previously described in Fertil et al. (2025). Briefly described here, the pendant drop tensiometer generates a droplet of solution (< 10 µL) suspended from a 0.9-mm diameter needle (Beier et al., 2019; Fertil et al., 2025). Droplets remain suspended for 300 s to reach equilibrium". What was the ambient relative humidity that the droplets exposed to in these measurements? When the droplets were at equilibrium with their surrounding, do the equilibrium composition of the droplets same as initial their stock solutions?

- Measurements were obtained at ambient room conditions; the temperature through the day ranged from 20.2-22C and RH ranged from 40-45% RH. Across all replicates presented, an average droplet volume change of 1.7% was observed between the start and end of measurement. This clarification has been added to the text:

  *"Surface tension measurements were run in triplicate; prior to each measurement, the tensiometer was flushed with DI water and ~ 2 mL of solution. Measurements were obtained at ambient room conditions, with temperature range of 20.2-22 °C and relative humidity range of 40-45 % RH."*

- Due to the short timeframes of suspension and this negligible change in droplet volume, it is not expected that evaporation effects will result in any meaningful change in droplet concentration between when the droplet is at equilibrium compared to its initial stock solution. The text has been modified to clarify this:

  *"Here, evaporation effects are negligible during the short suspension times. Therefore, the organic molar concentration C is equivalent to the droplet solution concentration as Eq. 2 can then be rearranged to solve for $D_S$ using dynamic surface tension measurements. ."*

Line 455, "Consequently, current hygroscopicity measurements that occur at fast time scales may not capture the full water uptake process of the synthesized organics and their mixtures". Can the authors comment if equilibrium hygroscopic measurements were achieved in all their investigated systems?

- We are unable to determine this as the residence time is so short for CCNC (10 seconds) and H-TDMA measurements (6.5 seconds)  but we hope the findings of this paper leads to more investigation regarding hygroscopic measurements of 2-MT/2-MTS at equilibrium and how to improve upon water uptake instrumentations. We acknowledge that this in our manuscript:

    *"Consequently, current hygroscopicity measurements that occur at fast time scales may not capture the full water uptake process of the synthesized organics and their mixtures. For example, the residence of aerosols within DMT CCNC columns is ~ 10 s (Paramonov et al., 2015) while similar H-TDMA instrument set ups have a residence time ~ 6.5 s (Mikhailov & Vlasenko, 2020). However, a previous study by Chuang et al. (2003) found atmospheric droplet growth timescales range between 5 to 100 s, congruent with the timescale of 2-MT and 2-MTS dynamic surface tension change (Fig. 2. and Chuang, 2003)."*

Other comments

For the water uptake measurements, would there be any volatility issue of 2-MT aerosols?

- In our CCNC experiments, we do not use large enough delta T's for 2-MT to volatilize during our runs. Additionally, if this process was occurring during both H-TDMA and CCNC experiments, we would see a significant change in kappa over the duration of the runs; this would be most apparent during the increase of SS from 0.4 to 1%. However, we do not observe this and our kappa values remain close to one another over multiple runs and supersaturations. In SI Table S24, the supersaturated kappa values remained between 0.1-0.2, which is a close range for kappa and constitute them as moderately hygroscopic according to Petters & Kreidenweis 2007.

References:

Petters, M. D., & Kreidenweis, S. M. (2007). A single parameter representation of hygroscopic growth and cloud condensation nucleus activity. *Atmos. Chem. Phys.*, *7*(8), 1961-1971. https://doi.org/10.5194/acp-7-1961-2007

Line 290, "However, in comparison to previously studied organics, 2-MT and 2-MTS σs/a remains close to pure water in the dilute bulk regime (Fig. 1). Thus, 2-MT and 2-MTS

surface activity is negligible for droplet activation." Can the authors also comment if droplet size would affect the results?

- Droplet size would not have a significant effect on these results as aerosol surface tension is dependent on surface area to volume ratio as opposed to droplet size alone (Bain et al., 2023). In theory, a higher concentrated organic solution (within a smaller droplet size) would result in greater surface tension depression. However, according to Bain et al., 2023 and Werner et al., 2025 aerosol surface tension would be much higher due to surface area-to-volume ratio dictating organic partitioning. As a result, aerosol droplets near activation would have surface tension values represented in the dilute bulk concentrations, as shown in Figure 1. As both 2-MT and 2-MTS surface tension remains close to water in this regime, aerosol surface tension would remain within water irrespective of droplet size.
- Additional text has been added in the manuscript to clarify this:

*"Previous studies by Bain et al. (2023) and Werner et al. (2025) emphasize the role of surface area-to-volume ratio dictating aerosol surface tension. Specifically, aerosol surface tension values are best represented by surface tension measurements of the organic in bulk solutions < 100 mM (Bain et al., 2023; Ferdousi-Rokib et al., 2025; Werner et al., 2025). Thus, 2-MT and 2-MTS surface activity is negligible for droplet activation as both dilute organic $\sigma_{s/a}$ is close to that of pure water (~72 mN m$^{-1}$)."*

References:

Bain, A., Ghosh, K., Prisle, N. L., & Bzdek, B. R. (2023). Surface-Area-to-Volume Ratio Determines Surface Tensions in Microscopic, Surfactant-Containing Droplets. *ACS Central Science*, *9*(11), 2076-2083. https://doi.org/10.1021/acscentsci.3c00998

Werner, E. K., Hammond, M., & Bain, A. (2025). Surface tension predictions during hygroscopic growth and cloud droplet activation using a simple kinetic surfactant partitioning model. *Aerosol Science and Technology*, *59*(7), 781–793. https://doi.org/10.1080/02786826.2025.2465705

Line 320, "This effect is more prominent in 2-MT than 2-MTS, as evident in its slower diffusion rates for concentrations >30 mM (Table S17)." Since the synthesized 2-MTS sample contain other species, will these species affect the diffusion rates?

- The other species may have an effect on the diffusion rates, as diffusion rates have been shown to be affected by composition in binary and higher order mixtures (Vignes 1966, Carrion et al., 2016). In our study, we observe how mixtures with AS can influence the diffusion rates. However, the influence of sodium methyl sulfate

(SMS) on diffusion rates is not as clear. Future research should focus on how SMS presence in mixtures can also influence diffusion rates via the methods presented in this study (e.g., dynamic surface tension measurements of 2-MTS sample/SMS mixtures).

- Additional clarification was added in the text:

*Both 2-MT and 2-MTS present complex viscous properties that may affect droplet phase and potentially change in the presence of inorganic compounds, such as AS. It is important to note that for 2-MTS, the remaining sample mass also contains SMS, which may further influence the estimated diffusion rates (Vignes, 1966; Guevara-Carrion et al., 2016). Future work should expand upon the methodology of this study to further understand the influence of SMS on viscous organic diffusivity, such as 2-MTS diffusion rates. Ultimately, diffusion effects were observed through dynamic surface tension measurements and may influence 2-MT, 2-MTS, and AS-mixed aerosol water uptake properties. Therefore, diffusion effects on synthesized organic and organic/AS aerosol mixtures were probed through water uptake measurements.*

References:

Vignes, A. (1966). Diffusion in Binary Solutions. Variation of Diffusion Coefficient with Composition. Industrial & Engineering Chemistry Fundamentals, 5(2), 189–199. https://doi.org/10.1021/i160018a007

Guevara-Carrion, G., Gaponenko, Y., Janzen, T., Vrabec, J., & Shevtsova, V. (2016). Diffusion in Multicomponent Liquids: From Microscopic to Macroscopic Scales. The Journal of Physical Chemistry B, 120(47), 12193–12210. https://doi.org/10.1021/acs.jpcb.6b09810

Line 345, "The organic 2-MT molecules do not diffuse fast enough to fully accumulate at the surface and substantially lower surface tension." As mentioned above, do the data collect at their equilibrium states?

- As mentioned in the previous response, volumes are held steady with an average ~1.7% volume change over the measurements. Thus, when the data is collected, the observed plateau we obtain in our surface tension measurements can be attributed to an equilibrium state as opposed to other effects such as evaporation.

Line 405, "Thus, it is believed that both 2-MT and 2-MTS organics slowly dissolve, and phase separate to form a relatively viscous phase under subsaturated conditions, corresponding to slow diffusion coefficients" When phase separation occurs, what is the morphology or thickness of the organic coating?

- The morphology is present as core-shell, observed in the AFM images for both 2-MT/AS and 2-MTS/AS aerosol and similar to other viscous organic aerosol studies (e.g., but not limited to Zhang et al., 2018, Song et. al., 2019, Gerrebos et al., 2024); the organic remains phase separated even as it is being slowly dissolved due to viscosity difference from the aqueous inorganic-containing phase. This has been further specified in the text:

*"A previous study by Cooke et al. (2022) observed a similar core-shell morphology for AS-seeded IEPOX-derived SOA particles; the study observed an organic shell, while the inorganic salt was observed to be present in the shell as well as within an aqueous core (Cooke et al., 2022). With AS dispersed on the outer shell as well as being present in an aqueous core, the inorganic salt in the shell will likely easily dissolve during water uptake and drive hygroscopicity, consistent with the results as observed in subsaturated hygroscopicity measurements."*

*"For 2-MT, the organic diffusion is limited under both sub- and supersaturated conditions, likely due to the undissolved viscous organic phase (Fig. 4A). Specifically, 2-MT viscosity causes slower dissolution compared to AS and results in the phase separated morphology."*

- As for thickness, it is difficult to quantify due to the flattening effects after particle impaction. Previous studies have modeled or estimated varied organic shell thickness. Examples include Schmedding et al., 2019 ranging thickness from 20 - 40 nm in their regional model and Riva et al., 2019 estimating an organic coating thickness of 40 nm for organosulfate/sulfate mixtures. We hope that future work will be able to utilize the synthesized 2-MT and 2-MTS samples to better estimate their organic shell thickness in organic-inorganic aerosol mixtures, as well as its implications for climate models.

References:

Zhang, Y., Chen, Y., Lambe, A. T., Olson, N. E., Lei, Z., Craig, R. L., Zhang, Z., Gold, A., Onasch, T. B., Jayne, J. T., Worsnop, D. R., Gaston, C. J., Thornton, J. A., Vizuete, W., Ault, A. P., & Surratt, J. D. (2018). Effect of the Aerosol-Phase State on Secondary Organic Aerosol Formation from the Reactive Uptake of Isoprene-Derived Epoxydiols (IEPOX). *Environmental Science & Technology Letters*, *5*(3), 167-174. https://doi.org/10.1021/acs.estlett.8b00044

Schmedding, R., Rasool, Q. Z., Zhang, Y., Pye, H. O. T., Zhang, H., Chen, Y., Surratt, J. D., Lee, B. H., Mohr, C., Lopez-Hilfiker, F. D., Thornton, J. A., Goldstein, A. H., and Vizuete, W.: Predicting Secondary Organic Aerosol Phase State and Viscosity and its Effect on Multiphase Chemistry in a Regional Scale Air Quality Model, Atmos. Chem. Phys. Discuss., https://doi.org/10.5194/acp-2019-900, 2019.

Riva, M., Chen, Y., Zhang, Y., Lei, Z., Olson, N. E., Boyer, H. C., Narayan, S., Yee, L. D., Green, H. S., Cui, T., Zhang, Z., Baumann, K., Fort, M., Edgerton, E., Budisulistiorini, S. H., Rose, C. A.,

Ribeiro, I. O., e Oliveira, R. L., dos Santos, E. O., Machado, C. M. D., Szopa, S., Zhao, Y., Alves, E. G., de Sá, S. S., Hu, W., Knipping, E. M., Shaw, S. L., Duvoisin Junior, S., de Souza, R. A. F., Palm, B. B., Jimenez, J.-L., Glasius, M., Goldstein, A. H., Pye, H. O. T., Gold, A., Turpin, B. J., Vizuete, W., Martin, S. T., Thornton, J. A., Dutcher, C. S., Ault, A. P., and Surratt, J. D.: Increasing Isoprene Epoxydiol-to-Inorganic Sulfate Aerosol Ratio Results in Extensive Conversion of Inorganic Sulfate to Organosulfur Forms: Implications for Aerosol Physicochemical Properties, Environ. Sci. Technol., 53, 8682–8694, https://doi.org/10.1021/acs.est.9b01019, 2019.

Song M, MacLean AM, Huang Y, Smith NR, Blair SL, et al. 2019.. Liquid-liquid phase separation and viscosity within secondary organic aerosol generated from diesel fuel vapors. . *Atmos. Chem. Phys.* 19:(19):12515–29

N. G. A. Gerrebos, J. Zaks, F. K. A. Gregson, M. Walton-Raaby, H. Meeres, I. Zigg, W. F. Zandberg and A. K. Bertram, High Viscosity and Two Phases Observed over a Range of Relative Humidities in Biomass Burning Organic Aerosol from Canadian Wildfires, *Environ. Sci. Technol.*, 2024, **58**, 21716–21728

For the AFM, can the authors comment how well the AFM represents the morphology airborne aqueous droplets?

We would like to acknowledge that AFM typically characterizes particles after deposition in under-saturated conditions, which may introduce morphological changes when compared with aerosols suspended as aqueous droplets. However, several studies listed below have shown that for aqueous droplets containing dissolved organics, the mixing state and morphology after evaporation can closely resemble that of the original airborne droplet, particularly when deposited onto inert or smooth surfaces like silica wafers. Given the controlled drying conditions and inert silica substrate used, we expect the AFM images to reasonably reflect the mixing state and morphology of the airborne droplets.

References:

Madawala, C. K., Lee, H. D., Kaluarachchi, C. P., & Tivanski, A. V. (2023). Quantifying the viscosity of individual submicrometer semisolid particles using atomic force microscopy. *Analytical chemistry*, *95*(39), 14566-14572.

Madawala, C. K., Molina, C., Kim, D., Gamage, D. K., Sun, M., Leibensperger III, R. J., ... & Tivanski, A. V. (2024). Effects of wind speed on size-dependent morphology and composition of sea spray aerosols. *ACS earth and space chemistry*, *8*(8), 1609-1622.

---

## Author Comment (AC2)

Reviewer #2:

We thank the reviewer for their comments. We have addressed each comment in a point by point basis

In principle, this is a useful experimental contribution to aerosol research. The main issue, though, is the purity of the 2-MTS is reported to be only 73 wt% pure. This is around 63 mole% based on the listed impurities. Therefore, it is difficult to know what to do with any of the measurements for 2-MTS, as they are only indicative of a mixed system and cannot be confidently assigned to the properties of pure 2-MTS.

Another issue is with the writing itself. There is an incredible overuse of the words traditionally (3 times) and traditional (16 times!). I would say those terms typically have no place in scientific writing. Especially if we're talking about kappa-Kohler theory and referencing a 2007 paper.

The word "traditional" was used as this kappa parameterization derives from the original kappa-Kohler theory; in recent papers, different versions of the kappa parameterization have been introduced (e.g., dependent on O/C, surface tension, Frenkel-Halsey-Hill theory, solubility) so the use of "traditional" was to differentiate it from these recent updated parameterizations (e.g., but not limited to Nakao et al., 2017, Gohil et al., 2022, Malek et al 2023) . We also wanted to differentiate the kappa values determined from Kohler theory and experimental kappa values.

For this manuscript, we have taken the reviewer's comments in mind and refined the use of the word "traditional."

References

Nakao, S. (2017). Why would apparent κ linearly change with O/C? Assessing the role of volatility, solubility, and surface activity of organic aerosols. Aerosol Science and Technology, 51(12), 1377–1388. https://doi.org/10.1080/02786826.2017.1352082

Gohil, K., Mao, C. N., Rastogi, D., Peng, C., Tang, M., & Asa-Awuku, A. (2022). Hybrid water adsorption and solubility partitioning for aerosol hygroscopicity and droplet growth. Atmos. Chem. Phys., 22(19), 12769–12787. https://doi.org/10.5194/acp-22-12769-2022

Malek, K., Gohil, K., Olonimoyo, E. A., Ferdousi-Rokib, N., Huang, Q., Pitta, K. R., Nandy, L., Voss, K. A., Raymond, T. M., Dutcher, D. D., Freedman, M. A., & Asa-Awuku, A. (2023). Liquid–Liquid Phase Separation Can Drive Aerosol Droplet Growth in Supersaturated Regimes. *ACS Environmental Au*. https://doi.org/10.1021/acsenvironau.3c00015

Also, overuse of scare quotes. E.g. salting-in is scare quoted two of three times in the text (but not in the title). Further, do "self-limiting" and "equilibrium" really need to be scare quoted? Both of those examples just confuse the reader. What are you even implying there?

These words were put in quotation marks initially to describe the definition of the term, for example:

*However, inorganic salts may also enhance organic dissolution, known as "salting in" (Riva et al., 2019).*

We do agree that quotation marks did not need to be used after as scare quotes, and have been removed.

Minor: show the structures for 2-MTS and MT in Figure 1 (you already show SOS and SDS). Citation: https://doi.org/10.5194/egusphere-2025-1935-RC2

We thank the reviewer for bringing this up - initially we only showed the SOS and SDS structures to demonstrate how the longer chained compounds resulted in decreased surface tension. However, we see how this may create confusion in the figure. Additionally, 2-MTS and MT can exist as different diastereomers and representing it on one figure may create further confusion (Chen et al., 2020. To simply Figure 1, we have removed the SOS and SDS structures.

Figure before:

[Figure]

Figure after:

[Figure]

References:
Clements, A. L., & Seinfeld, J. H. (2007). Detection and quantification of 2-methyltetrols in ambient aerosol in the southeastern United States. Atmospheric Environment, 41(9), 1825–1830. https://doi.org/https://doi.org/10.1016/j.atmosenv.2006.10.056

Chen, Y., Zhang, Y., Lambe, A. T., Xu, R., Lei, Z., Olson, N. E., Zhang, Z., Szalkowski, T., Cui, T., Vizuete, W., Gold, A., Turpin, B. J., Ault, A. P., Chan, M. N., & Surratt, J. D. (2020). Heterogeneous Hydroxyl Radical Oxidation of Isoprene-Epoxydiol-Derived Methyltetrol Sulfates: Plausible Formation Mechanisms of Previously Unexplained Organosulfates in Ambient Fine Aerosols. Environmental Science & Technology Letters, 7(7), 460–468. https://doi.org/10.1021/acs.estlett.0c00276

---

## Author Response (AR2)

**Dear Editor Knopf,**

Thank you for clarifying the reviewers' comments and the missing components of our response. We have gone through the manuscript and have added more clarity regarding the unknown role of the impurities (SMS specifically) in our measurements. We have listed the changes made in the manuscript below and have addressed Reviewer #2's concerns with greater details:

**Specifically, the following changes have been made -**

Lines 390-397: It should be noted that the synthesized 2-MTS sample is 73% pure 2-MTS and is likely mixed with AS and SMS. Both SMS and AS (Fig.1, red circles; Table S16) have surface tension values, > 72 mN m-1 in the dilute regime. However, despite the presence of impurities in the mixture, synthesized 2-MTS surface tension reaches values ~ 68 mN m-1. Therefore, the presence of these impurities may counteract possible further surface tension depression exhibited by 2-MTS. Future work can better probe surface tension of the pure organic 2-MTS and effects of SMS by applying a multicomponent surface tension model (e.g., multicomponent models of Topping et al., 2007) to dynamic surface tension measurements.

**Figure 2 Caption now states:**

Dynamic  $\sigma$ s/a measurements for (A) 3-94 mM 2-MT sample/500 mM AS and (B) 3-53 mM 2-MTS sample/500 mM AS mixtures. Dynamic  $\sigma$ s/a was recorded over a duration of 300 seconds. The 2-MTS sample mixtures contain additional AS (3 wt%) and SMS (24 wt%) due to impurities, which may further influence dynamic surface tension measurements.

**Line 472-481**

It is important to note that for 2-MTS, the remaining sample mass also contains SMS, which may further influence the estimated diffusion rates (Vignes, 1966; Wallace et al., 2021). Diffusion coefficients within aerosols may be sensitive to mixture ratio, as observed by Wallace et al. (2021). Thus, the presence of SMS may be affecting the 2-MTS sample/AS diffusion rates observed in this study. Future work should explore the influence of SMS on viscous organic diffusivity by applying this study's methodology to a range of 2-MTS sample/SMS mixtures. Ultimately, diffusion effects were observed through dynamic surface tension measurements and may influence 2-MT, 2-MTS, and AS-mixed aerosol water uptake properties. Therefore, diffusion

effects on synthesized organic and organic/AS aerosol mixtures were probed through water uptake measurements.

**Line 491-497:**

For supersaturated hygroscopicity, the CCNC instrument setup was used to obtain experimental  $D_{p,50}$  values across multiple supersaturation conditions (0.31, 0.43, 0.65, 0.88, 1.10, 1.32, and 1.54 % SS); the critical diameter values for 2-MT/AS and 2-MTS/AS mixtures are listed in Tables S22-S23. For 100 wt% 2-MTS hygroscopicity, impurity (SMS and additional AS) hygroscopicity are accounted for by applying ZSR mixing rule (Eq. 6) to solve for pure organic hygroscopicity; SMS  $\kappa$  was assumed to be  $\sim$ 0.459 based on Peng et al. (2021).

**Figure 3 caption states:**

**Figure 1.** Experimental κ-hygroscopicity measurements derived from (A) H-TDMA measurements and (B) CCNC measurements. 2-MTS/AS sample mixture wt% was adjusted based on the presence of AS and SMS impurities (Table S21). Subsaturated hygroscopicity (κH-TDMA) of 2-MT/AS and 2-MTS /AS mixtures are represented as open orange squares and open purple circles, respectively. Supersaturated hygroscopicity (κCCN) for 2-MT/AS and 2-MTS sample/AS mixtures are represented as orange squares and purple circles, respectively. For 100 wt% 2-MTS, κ values were adjusted to account for impurities by applying mixing rule, assuming an SMS κ of ~0.459 and AS κ ~ 0.61 (Eq. 6, Table S20). κ-Köhler theory (κZSR) was used to predict hygroscopicity of 2-MT/AS (solid orange line) and 2-MTS sample/AS (solid purple line) via Eq. 6. Organic κint was determined from 100 wt% κCCN. 2-MT κint (yellow dashed line) was determined to be 0.269. 2-MTS κint (blue dashed line) was determined to be 0.139.

**Line 525 - 531**

For 2-MTS/AS mixtures, both subsaturated and supersaturated hygroscopic trends may be further impacted by the presence by SMS. Impurity hygroscopicity in 100 wt% 2-MTS  $\kappa$  was accounted for by using ZSR mixing rule, which assumes ideal interactions between SMS, AS, and 2-MTS. However, non-idealities (e.g., phase separation, salting in) may result in SMS having a greater influence on hygroscopicity. Future studies can examine 2-MTS sample hygroscopicity in a ternary system (e.g., ternary system of Malek et al., 2023) with AS and SMS better clarify the influence of both compounds on 2-MTS hygroscopicity.

**Line 679-685:**

As a result, the viscous organic phase remains intact while aqueous AS in the core drives hygroscopicity. A caveat to these results is the presence of SMS, an organosulfate, being present within the 2-MTS sample at ~24 wt%. Therefore, SMS may have an effect on surface tension, diffusivity, and hygroscopic trends observed for the 2-MTS sample/AS mixtures that is currently unknown in this study. Future work may utilize the methodology laid out in this work to better understand the influence of SMS and any additional mixture component on viscous organic properties and water uptake.

We have also shortened the abstract to be under 250 words. We thank you and the reviewers for your efforts.

Sincerely,

Ferdousi-Rokib et al.